# A Reduction from Delayed to Immediate Feedback for Online Convex Optimization with Improved Guarantees

## Abstract

We develop a reduction-based framework for online learning with delayed feedback that recovers and improves upon existing results for both first-order and bandit convex optimization. Our approach introduces a continuous-time model under which regret decomposes into a delay-independent learning term and a delay-induced drift term, yielding a delay-adaptive reduction that converts any algorithm for online linear optimization into one that handles arbitrary delays. For bandit convex optimization, we significantly improve existing regret bounds, with delay-dependent terms matching state-of-the-art first-order rates. For first-order feedback, we recover state-of-the-art regret bounds via a simpler, unified analysis.

Quantitatively, for bandit convex optimization we obtain $O(\sqrt{d_{\text{tot}}} + T^{\frac{3}{4}}\sqrt{k})$ regret, improving the delay-dependent term from $O(\min\{\sqrt{Td_{\max}}, (Td_{\text{tot}})^{\frac{1}{3}}\})$ in previous work to $O(\sqrt{d_{\text{tot}}})$. Here, $k$, $T$, $d_{\max}$, and $d_{\text{tot}}$ denote the dimension, time horizon, maximum delay, and total delay, respectively. Under strong convexity, we achieve $O(\min\{\sigma_{\max}\ln T, \sqrt{d_{\text{tot}}}\} + (T^2 \ln T)^{\frac{1}{3}}k^{\frac{2}{3}})$, improving the delay-dependent term from $O(d_{\max}\ln T)$ in previous work to $O(\min\{\sigma_{\max}\ln T, \sqrt{d_{\text{tot}}}\})$, where $\sigma_{\max}$ denotes the maximum number of outstanding observations and may be considerably smaller than $d_{\max}$.

## 1. Introduction

Online convex optimization (OCO) is a fundamental framework for sequential decision-making under uncertainty (Orabona, 2025; Hazan, 2023). In the classic OCO setting, a player repeatedly selects an action from a convex domain, incurs a loss determined by a convex loss function, and observes some feedback about this loss function. In the full-information setting, the player is given the loss function. In the first-order setting, the player observes the gradient of the loss function at the chosen action. In the zeroth-order (bandit) setting, only the scalar loss value is revealed.

In practice, observations are often *delayed*: feedback from round $t$ may arrive only at round $t + d_t$, where $d_t$ is the round-dependent delay. This forces the player to make decisions without up-to-date information. For example, consider a recommendation system that must select content instantaneously for each arriving user. Feedback needed to update the system's parameters (such as clicks or purchases) arrives only after user sessions conclude. When many users are served concurrently, new arrivals must be handled while feedback from earlier users is pending, resulting in delays. Delayed feedback has been studied extensively in many variants of sequential decision-making, including online learning, bandits, and reinforcement learning (e.g., Mesterharm, 2005; Cesa-Bianchi et al., 2016; Zimmert & Seldin, 2020; van der Hoeven et al., 2023).

A natural question is whether delayed algorithms can be obtained as reductions from non-delayed ones. Prior work has made significant progress on this question (Weinberger & Ordentlich, 2002; Joulani et al., 2013; 2016). However, existing approaches either require advance knowledge of delays, incur memory overhead scaling with the delay, or are limited to full-information feedback. Developing a unified, delay-adaptive reduction—one that requires no advance knowledge of delays, adapts to function regularity (e.g., strong convexity), and incorporates different feedback models (first-order or bandit)—is the main goal of this work.

In this paper, we introduce a continuous-time model for delayed feedback that subsumes the standard discrete-time model and enables a clean reduction from delayed feedback to immediate feedback for both online convex optimization with first-order feedback (OCO) and bandit convex optimization (BCO). For BCO, this reduction yields improved regret rates, while for OCO it recovers state-of-the-art bounds via a simple, unified analysis. Our contributions are summarized as follows:

[1]Anonymous Institution, Anonymous City, Anonymous Region, Anonymous Country. Correspondence to: Anonymous Author <anon.email@domain.com>.

Preliminary work. Under review by the International Conference on Machine Learning (ICML). Do not distribute.

*Table 1.* Notation: $T$ is the horizon, $k$ is the dimension, $d_{\text{tot}} = \sum_{t=1}^{T} d_t$ is the total delay, $d_{\max} = \max_{t \in [T]} d_t$ is the maximum delay, and $\sigma_{\max} \leqslant d_{\max}$ is the maximum number of outstanding observations at any round. The bounds of Wan et al. (2024) require advance knowledge of $d_{\max}$ and $T$; all other algorithms are fully adaptive.

| Loss type | Regret bounds for bandit convex optimization with delays | | |
|---|---|---|---|
| | (Bistritz et al., 2022) | (Wan et al., 2024) | Our work |
| Convex | $(Td_{\text{tot}})^{1/3}\sqrt{k} + T^{3/4}k$ | $\sqrt{Td_{\max}} + T^{3/4}\sqrt{k}$ | $\sqrt{d_{\text{tot}}} + T^{3/4}\sqrt{k}$ |
| Strongly convex | N/A | $d_{\max} \ln T + (T^2 \ln T)^{1/3}k^{2/3}$ | $\min\{\sigma_{\max} \ln T, \sqrt{d_{\text{tot}}}\} + (T^2 \ln T)^{1/3}k^{2/3}$ |

1. **Continuous-time model (Section 3).** We introduce a continuous-time model for delayed feedback. Rather than the usual round-based view where each round reveals an unordered batch of observations, our model places predictions and observations as events on a shared timeline; the delay structure then emerges from their interleaving. Theorem 3.4 characterizes this structure, identifying new invariants and equivalences among delay-related quantities that appear throughout the prior delayed-learning literature. Ordering the interaction by observation times provides a natural framework for analyzing algorithms that run single-instance updates upon each feedback arrival (e.g., Joulani et al. (2016)), yielding a regret decomposition into a non-delayed term and a prediction-drift term (Figure 4). Our analysis provides new tools for controlling this drift, which we exploit to improve regret bounds through a unified analysis, without requiring advance knowledge of problem parameters such as the maximum delay $d_{\max}$, total delay $d_{\text{tot}}$, maximum number of outstanding observations $\sigma_{\max}$, or the time horizon $T$.

2. **Reduction to non-delayed learning (Sections 4–5).** Building on this decomposition, we reduce delayed OCO to online linear optimization with penalties for changing predictions, a problem with no delays. We provide wrappers that transform any algorithm for this setting into a delayed OCO or BCO algorithm whose regret is controlled by the base algorithm's performance (Theorems 5.2 and 5.6). The BCO wrapper incorporates single-point gradient estimation (Flaxman et al., 2004).

3. **First-order feedback (Section 5.1).** Wrapping Proximal Follow-The-Regularized-Leader (P-FTRL) and Online Mirror Descent (OMD) update rules recovers state-of-the-art bounds for delayed OCO: $O(\sqrt{T + d_{\text{tot}}})$ for convex losses (Corollary 5.3) and $O(\min\{\sigma_{\max} \ln T, \sqrt{d_{\text{tot}}}\} + \ln T)$ for strongly convex losses (Corollary 5.4), matching Quanrud & Khashabi (2015) and Qiu et al. (2025) respectively via an adaptive approach requiring no knowledge of delays or horizon.

4. **Bandit feedback (Section 5.2).** For delayed BCO, wrapping P-FTRL achieves $O(\sqrt{d_{\text{tot}}} + T^{3/4}\sqrt{k})$ expected regret (Corollary 5.7), where $k$ is the dimen-

sion of the domain, improving the delay dependence over $(Td_{\text{tot}})^{1/3}$ of Bistritz et al. (2022) and $\sqrt{Td_{\max}}$ of Wan et al. (2024). For strongly convex losses, we obtain $O(\min\{\sigma_{\max} \ln T, \sqrt{d_{\text{tot}}}\} + (T^2 \ln T)^{1/3}k^{2/3})$ regret (Corollary 5.8), improving over $O(d_{\max} \ln T + (T^2 \ln T)^{1/3}k^{2/3})$ of Wan et al. (2024). Note that $\sigma_{\max}$ may be considerably smaller than $d_{\max}$. These results are summarized in Table 1.

5. **Skipping scheme (Section 6).** We incorporate the adaptive skipping technique of Zimmert & Seldin (2020), as an external wrapper (Algorithm 3), improving the delay term from $O(\sqrt{d_{\text{tot}}})$ to $O(\min_Q\{|Q| + \sqrt{\sum_{t \notin Q} d_t}\})$. This significantly improves regret when a few rounds have exceptionally large delays (Corollary 6.2).

**Related work.** Methodologically, the work closest to ours is Joulani et al. (2016), who study delayed OCO with full-information feedback. They propose a black-box-style reduction that feeds observations to a base (non-delayed) algorithm in order of arrival, bounding regret in terms of the base algorithm's regret and prediction drift. With known $d_{\text{tot}}$, they achieve regret $O(\sqrt{T + d_{\text{tot}}})$. They also establish a bound of $O(\sqrt{T\tau^*})$, where $\tau^*$ is the maximum number of observations that can arrive while some earlier feedback is still outstanding, known in advance. Using the tools developed in this paper, we show that $\tau^* = \Theta(d_{\max})$, thus expressing this result in standard notation as $O(\sqrt{Td_{\max}})$

Weinberger & Ordentlich (2002) first studied online learning with delayed feedback and proposed a reduction that runs $d_{\text{fixed}} + 1$ parallel copies of a base non-delayed algorithm, yielding regret $d_{\text{fixed}} \cdot R(T/d_{\text{fixed}})$ for a fixed, known delay $d_{\text{fixed}}$, where $R(T)$ denotes the base regret. Joulani et al. (2013) extended this to round-dependent delays, giving $d_{\max} \cdot R(T/d_{\max})$. Quanrud & Khashabi (2015) showed that, in delayed OCO, a simple delayed gradient-descent scheme, updating each round using whatever gradients have arrived so far with stepsize $\eta \asymp 1/\sqrt{T + d_{\text{tot}}}$, achieves regret $O(\sqrt{T + d_{\text{tot}}})$. Delayed OCO with strongly convex losses has also received attention (Wan et al., 2021; Wu et al., 2024; Qiu et al., 2025). Héliou et al. (2020) first extended the bandit gradient descent method of Flaxman et al. (2004) to delayed BCO. Bistritz et al. (2022) developed the

first fully delay-adaptive algorithm using a doubling trick with respect to both horizon and total delay, and Wan et al. (2024) improved the bound via a blocking strategy; however, their method is not delay-adaptive, requiring knowledge of $d_{\max}$ and $T$.

## 2. Problem Setting

Let $T \in \mathbb{N}$ denote the time horizon, and let $\mathcal{K} \subset \mathbb{R}^k$ be a non-empty, convex, and closed domain for the dimension $k \in \mathbb{N}$. We equip $\mathbb{R}^k$ with a norm $\|.\|$ and its dual norm $\|.\|_\star$.

The player interacts with an environment determined by an oblivious adversary over $T$ rounds (Fig. 1). Before the game begins, the adversary selects delays $d_t$ and convex, differentiable loss functions $f_t : \mathcal{K} \to \mathbb{R}$ for each round. Every round $t$, the player predicts $x_t \in \mathcal{K}$, receiving the corresponding feedback (gradient $\nabla f_t(x_t)$ or value $f_t(x_t)$, depending on the feedback model) at the end of round $t + d_t$. We assume $d_t \in [T - t]$ for all $t \in [T]$, since feedback arriving after round $T$ cannot affect any prediction.

---

**Online Convex Optimization with Delays**

- *Latent parameters:* number of rounds $T$.
- *Pre-game:* adversary selects loss functions $f_t : \mathcal{K} \to \mathbb{R}$ and delays $d_t \in [T - t]$ for $t \in [T]$.

For each round $t = 1, 2, \ldots, T$:
1. The player predicts $x_t \in \mathcal{K}$, incurring loss $f_t(x_t)$.
2. For $s$ such that $s + d_s = t$, the environment reveals:
   - $(s, \nabla f_s(x_s))$ in the first-order feedback model,
   - $(s, f_s(x_s))$ in the bandit feedback model.

---

*Figure 1.* OCO with delays under first-order or bandit feedback.

The regret against a comparator $u \in \mathcal{K}$ is defined as $R_T(u) = \sum_{t=1}^{T}(f_t(x_t) - f_t(u))$, and its expected regret as $\bar{R}_T(u) = \mathbb{E}[R_T(u)]$. Letting $x^* \in \arg\min_{x \in \mathcal{K}} \sum_{t=1}^{T} f_t(x)$ denote the best action in hindsight, the objective is to minimize $R_T(x^*)$ almost surely or its expectation $\bar{R}_T(x^*)$.

**Regularity assumptions.** We impose the following standard regularity assumptions, which are necessary to obtain sublinear regret guarantees in online convex optimization.

**Assumption 2.1.** The diameter of $\mathcal{K}$ is bounded by $D$, i.e., $\sup_{x,y \in \mathcal{K}} \|x - y\| \leqslant D$.

**Assumption 2.2.** For all $t \in [T]$, $f_t$ has norm of its gradient bounded by $G \geqslant 0$, i.e., $\sup_{x \in \mathcal{K}} \|\nabla f_t(x)\|_\star \leqslant G$.

Assumptions 2.1 and 2.2 hold throughout the paper. When deriving stronger guarantees, we also impose the following strong convexity condition, stated explicitly when invoked.

**Assumption 2.3.** For all $t \in [T]$, $f_t$ is $\lambda$-strongly convex for $\lambda > 0$, i.e., $f_t(y) \geqslant f_t(x) + \langle \nabla f_t(x), y - x \rangle + \frac{\lambda}{2} \|y - x\|^2$

for all $x, y \in \mathcal{K}$.

For bandit convex optimization, we take $\|\cdot\|$ to be Euclidean norm $\|\cdot\|_2$ and impose two additional assumptions.

**Assumption 2.4.** For all $t \in [T]$, absolute values of $f_t$ are bounded by $M \geqslant 0$, i.e., $\sup_{x \in \mathcal{K}} |f_t(x)| \leqslant M$.

**Assumption 2.5.** The domain satisfies $r\mathbb{B}^k \subseteq \mathcal{K} \subseteq R\mathbb{B}^k$ for some radii $0 < r < R$.

*Remark* 2.6 (Differentiability of loss functions). While we assume throughout that the loss functions $f_t$ are differentiable, weaker regularity suffices: for first-order feedback, subdifferentiability with subgradient norms bounded by $G$, and for bandit feedback, integrability with $G$-Lipschitzness. We do not pursue this generality as it does not materially affect the analysis or the resulting bounds.

**Steady algorithms.** An algorithm in this setting is a (possibly randomized) mapping from the history of received feedback and information about delays to predictions. The class of *steady algorithms* is formally defined below.

**Definition 2.7** (Steady Algorithm). An algorithm for online learning with delays is *steady* if it does not change its prediction after rounds without feedback: $x_t = x_{t+1}$ whenever $\{s : s + d_s = t\} = \varnothing$. It is *steady in expectation* if $\mathbb{E}[x_t] = \mathbb{E}[x_{t+1}]$ whenever $\{s : s + d_s = t\} = \varnothing$.

**Notation.** For $n \in \mathbb{N}$, we let $[n] = \{1, \ldots, n\}$. For any sequence $\{a_t\}_{t=1}^{T}$, we write $a_{\text{tot}} = \sum_{t=1}^{T} a_t$ and $a_{\max} = \max_{t \in [T]} a_t$. We denote by $\mathbb{B}^k$ and $\mathbb{S}^{k-1}$ the unit Euclidean ball and sphere centered at the origin in $\mathbb{R}^k$, respectively. For any $S \subseteq \mathbb{R}^k$ and $c \in \mathbb{R}$, we let $cS = \{cx : x \in S\}$.

## 3. Continuous Time Model for Delays

The classical formulation of online learning with delays assumes a discrete, round-based time structure: a player makes a prediction in round $t$ and receives feedback in round $t + d_t$. Real-world systems, however, generate predictions and observations asynchronously. Rather than forcing these events into discrete rounds, we adopt a continuous time model that places them on a timeline $\mathbb{R}_+$.

**Definition 3.1** (Continuous Time Model (CTM)). The timing of predictions and observations is determined by latency-intervals $\{I_t\}_{t=1}^{T}$, where $I_t = (l_t, r_t) \subset \mathbb{R}_+$. For every $t \in [T]$, the environment requests a prediction at time $l_t$ and reveals the corresponding observation at time $r_t$. All timestamps $\{l_t, r_t\}_{t=1}^{T}$ are distinct, and intervals are indexed in increasing order of prediction times: $0 < l_1 < \ldots < l_T$.

In the round-based model, the delay sequence $\{d_t\}_{t=1}^{T}$ fully specifies the time structure. In particular, it determines the number of outstanding observations in each round, $\sigma_t = \sum_{s=1}^{t-1} \mathbb{I}(s + d_s \geqslant t)$, which is commonly used in

the literature; for brevity, we refer to $\sigma_t$ as the *backlog* in this paper. Under the CTM, both delays and backlogs arise directly from the geometry of the latency intervals.

Section 3.1 formalizes this structure. Building on it, Section 3.2 introduces an observation-centric view of the problem that complements the standard prediction-centric perspective. We show that steady algorithms admit a natural regret decomposition: non-delayed regret on observation-ordered losses plus delay-induced prediction drift (Fig. 4).

### 3.1. Delays, Backlogs, and Their Duals

Under the CTM, rounds of the delayed problem (Fig. 1) naturally correspond to time-intervals: round $t$ spans the interval $[l_t, l_{t+1})$, during which the $t$-th prediction is made at time $l_t$, and observations arrive at times $\{r_s : r_s \in (l_t, l_{t+1})\}$. Therefore, the discrete *delay* $d_t$ and *backlog* $\sigma_t$ arise as

$$d_t = |\{s : l_s \in I_t\}| \quad \text{and} \quad \sigma_t = |\{s : l_t \in I_s\}|, \quad (1)$$

where $d_t$ counts how many predictions are made between the $t$-th prediction and observing its feedback, and $\sigma_t$ counts how many previous predictions still await feedback immediately before making the $t$-th prediction.

The following lemma and remark show that these definitions recover the classical round-based formulation.

**Lemma 3.2.** *For quantities in* (1)*, it holds that:*

$$r_t \in (l_{t+d_t}, l_{t+d_t+1}), \quad \sigma_t = |\{s : s < t \leqslant s + d_s\}|.$$

*Proof.* Exactly $|\{s : l_s < r_t\}| = |\{s : l_s \leqslant l_t\}| + |\{s : l_t < l_s < r_t\}| = t + d_t$ prediction times precede time $r_t$, so $r_t \in (l_{t+d_t}, l_{t+d_t+1})$. Then, the identity for backlog $\sigma_t = |\{s : l_s < l_t < r_s\}|$ follows immediately. $\square$

*Remark* 3.3 (Consistency and Realizability in the CTM). Lemma 3.2 confirms that our definitions are consistent with round-based protocols: $d_t$ equals the number of round-intervals $[l_s, l_{s+1})$ spanned by $I_t$, placing observation time $r_t$ in round $t + d_t$; $\sigma_t$ matches the classical definition as the number of outstanding delays. Conversely, any round-based delayed-feedback protocol can be represented within the CTM. Specifically, given a delay sequence $\{d_t\}_{t=1}^T$, we can set $(l_t, r_t) = (t, t + d_t + 1 - 2^{-t})$ so that round $t$ spans interval $[t, t+1)$ and observation time $r_t$ clearly falls within round $t + d_t$ spanned by interval $[t + d_t, t + d_t + 1)$.

To fully characterize the delay structure in the CTM, we define novel *dual-delay* and *dual-backlog* sequences

$$d_t^\star = |\{s : r_s \in I_t\}| \quad \text{and} \quad \sigma_t^\star = |\{s : r_t \in I_s\}|. \quad (2)$$

Here, dual-delay $d_t^\star$ counts how many observations occur between the $t$-th prediction and its observation, while $\sigma_t^\star$

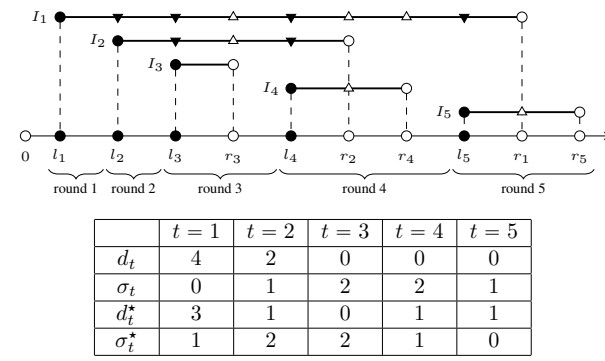

| | $t=1$ | $t=2$ | $t=3$ | $t=4$ | $t=5$ |
|---|---|---|---|---|---|
| $d_t$ | 4 | 2 | 0 | 0 | 0 |
| $\sigma_t$ | 0 | 1 | 2 | 2 | 1 |
| $d_t^\star$ | 3 | 1 | 0 | 1 | 1 |
| $\sigma_t^\star$ | 1 | 2 | 2 | 1 | 0 |
| $\beta(t)$ | 4 | 2 | 1 | 3 | 5 |

*Figure 2.* Example of latency-intervals for $T = 5$. The table provides values $d_t, \sigma_t, d_t^\star, \sigma_t^\star$ and $\beta : [T] \to [T]$ from Theorem 3.4.

counts how many predictions still await feedback when receiving observation for the $t$-th prediction.

The following theorem establishes the fundamental properties and relations among the four sequences.

**Theorem 3.4.** *For quantities in* (1) *and* (2)*, it holds that:*

(a) $\sum_{t=1}^T d_t = \sum_{t=1}^T \sigma_t = \sum_{t=1}^T d_t^\star = \sum_{t=1}^T \sigma_t^\star$,

(b) $\sigma_{\max} = \sigma_{\max}^\star$, $\frac{1}{2}d_{\max} \leqslant d_{\max}^\star \leqslant 2d_{\max}$,

(c) $d_t^\star = \sigma_t + \beta(t) - t$, $\sigma_t^\star = d_t + t - \beta(t)$,

*where* $\beta : [T] \to [T]$ *is the permutation* $\beta(t) = |\{s : r_s \leqslant r_t\}|$.

In particular, all four sequences sum to the total delay $d_{\text{tot}}$. Figure 2 illustrates this in a concrete example: delays $d_t$ (dual-delays $d_t^\star$) correspond to black (white) triangles along intervals $I_t$, while backlogs $\sigma_t$ (dual-backlogs $\sigma_t^\star$) correspond to black (white) triangles at timestamps $l_t$ ($r_t$), with the total number of triangles of each color $d_{\text{tot}} = 6$.

More properties of the CTM and their proofs are provided in Appendix B.

### 3.2. Observation-Ordering and Steady Algorithms

The CTM naturally suggests an observation-centric perspective, in which we order quantities by observation times rather than by prediction times. This viewpoint complements the standard prediction-centric view in the literature and is essential for our analysis of steady algorithms. To facilitate this, we introduce the notational convention for *observation-ordering*, which we adopt throughout the paper.

**Definition 3.5** (Observation-Ordering). Under the CTM, let $\rho : [T] \to [T]$ be the permutation such that $r_{\rho(1)} < ... < r_{\rho(T)}$. For any sequence $\{a_t\}_{t=1}^T$ indexed by $t \in [T]$, define its observation-ordering $\{\tilde{a}_n\}_{n=1}^T$ by $\tilde{a}_n = a_{\rho(n)}$ for $n \in [T]$.

For ease of notation, we use subscripts $n \in [T]$ for observation-ordered quantities and $t \in [T]$ for prediction-ordered quantities throughout the paper.

*Figure 3.* Partition of rounds induced by the latency intervals in Fig. 2: if $t \in Z_n$ (equivalently, $l_t \in [\widetilde{r}_{n-1}, \widetilde{r}_n)$), then a steady algorithm outputs $x_t = z_n$. The table provides corresponding values $\widetilde{d}_n, \widetilde{\sigma}_n, \widetilde{d}_n^\star, \widetilde{\sigma}_n^\star$ and permutation $\rho : [T] \to [T]$.

Under observation-ordering, the $n$-th observation arrives at time $\widetilde{r}_n$ for the prediction $\widetilde{x}_n$ made at time $\widetilde{l}_n$ and evaluated on loss function $\widetilde{f}_n$. Setting $\widetilde{r}_0 = 0$, we obtain a chronological sequence of observation times $0 = \widetilde{r}_0 < \widetilde{r}_1 < \ldots < \widetilde{r}_T$ such that all timestamps lie within $[\widetilde{r}_0, \widetilde{r}_T]$. Since no two timestamps coincide, each prediction time $\widetilde{l}_n$ lies strictly between two consecutive observation times $\{\widetilde{r}_n\}_{n=0}^T$. The following lemma, the observation-centric analogue of Lemma 3.2, characterizes this interleaving and expresses the dual-backlog $\widetilde{\sigma}_n^\star$ as the number of outstanding dual-delays.

**Lemma 3.6.** *For quantities in* (2), *it holds that:*

$$\widetilde{l}_n \in (\widetilde{r}_{n - \widetilde{d}_n^\star - 1}, \widetilde{r}_{n - \widetilde{d}_n^\star}), \quad \widetilde{\sigma}_n^\star = |\{m : m - \widetilde{d}_m^\star \leqslant n < m\}|.$$

*Proof.* Exactly $|\{m : \widetilde{r}_m < \widetilde{l}_n\}| = |\{m : \widetilde{r}_m < \widetilde{r}_n\}| - |\{m : \widetilde{l}_n < \widetilde{r}_m < \widetilde{r}_n\}| = n - 1 - \widetilde{d}_n^\star$ observations precede time $\widetilde{l}_n$, so $\widetilde{l}_n \in (\widetilde{r}_{n - \widetilde{d}_n^\star - 1}, \widetilde{r}_{n - \widetilde{d}_n^\star})$. The identity for $\widetilde{\sigma}_n^\star = |\{m : \widetilde{l}_m < \widetilde{r}_n < \widetilde{r}_m\}|$ follows immediately. $\quad\square$

The observation-centric perspective is particularly useful for analyzing steady algorithms (Definition 2.7). Since a steady algorithm does not change its prediction in rounds without feedback, it admits a *base-prediction sequence* $\{z_n\}_{n=1}^T \subset \mathcal{K}$ such that the prediction remains constant between consecutive observation times: for each $n \in [T]$ and every prediction time $l_t \in (\widetilde{r}_{n-1}, \widetilde{r}_n)$, the algorithm outputs $x_t = z_n$. Equivalently, the rounds $[T]$ partition into sets $Z_n = \{t : l_t \in [\widetilde{r}_{n-1}, \widetilde{r}_n)\}$, with $x_t = z_n$ if and only if $t \in Z_n$. Figure 3 illustrates this for the example in Figure 2.

Lemma 3.6 links the observation-ordered predictions to the base-predictions through the dual-delays: $\widetilde{x}_n = z_{n - \widetilde{d}_n^\star}$. Substituting this identity, we can write the regret of any steady algorithm entirely in terms of the base sequence $z_n$ evaluated on the observation-ordered losses $\widetilde{f}_n$. For $G$-Lipschitz losses, this yields the following decomposition:

$$\boxed{\begin{aligned} R_T(x^*) &= \sum_{n=1}^T \big(\widetilde{f}_n(z_{n - \widetilde{d}_n^\star}) - \widetilde{f}_n(x^*)\big) \\ &\leqslant \underbrace{\sum_{n=1}^T \big(\widetilde{f}_n(z_n) - \widetilde{f}_n(x^*)\big)}_{\text{Non-Delayed Regret}} + G \underbrace{\sum_{n=1}^T \|z_n - z_{n - \widetilde{d}_n^\star}\|}_{\text{Prediction Drift}}. \end{aligned}}$$

*Figure 4.* Regret decomposition for steady algorithms.

Here, the *non-delayed regret* is the regret of the sequence $\{z_n\}_{n=1}^T$ on the observation-ordered losses $\{\widetilde{f}_n\}_{n=1}^T$ and the *prediction drift* captures the cost of the played predictions $\widetilde{x}_n = z_{n - \widetilde{d}_n^\star}$ lagging behind base-predictions $z_n$.

This decomposition underlies our reduction from online learning with delays to online learning with drift-penalization, which we define in the following section.

## 4. Online Learning with Drift Penalization

This section introduces online linear optimization with drift penalization (Figure 5), a game that serves as the foundation for reductions in subsequent sections. Crucially, this formulation involves no delays; the connection to delayed feedback emerges only through the reductions.

In this game, a player selects predictions from $\mathcal{K}$ (the same domain as in OCO with delays) over $N$ rounds, incurring in each round a linear loss and a penalty for moving too far from past predictions, measured against one of them. Specifically, in round $n$, the player chooses $z_n$ and incurs the loss $\langle c_n, z_n \rangle$ plus the drift penalty $\|z_n - z_{n - \lambda_n}\|$, specified by the lag $\lambda_n$. The loss vectors and lags $\{c_n, \lambda_n\}_{n=1}^N$ are fixed by an adversary before the game begins. To update his strategy, after each round the player observes $(c_n, \lambda_n)$ as well as the number of outstanding lags $\nu_n = |\{m : m - \lambda_m \leqslant n < m\}|$, which summarizes how many future penalties will involve predictions up to the current round $n$.

---

**Online Linear Optimization with Drift Penalization**

- *Latent parameters:* number of rounds $N$.
- *Pre-game:* adversary picks loss vectors $c_n \in \mathbb{R}^k$ and lags $\lambda_n \in \{0, \ldots, n-1\}$ for $n \in [N]$.

For each round $n = 1, 2, \ldots, N$:
1. The player predicts $z_n \in \mathcal{K}$, incurring loss $\langle c_n, z_n \rangle$ and drift penalty $\|z_{n - \lambda_n} - z_n\|$.
2. The environment reveals a tuple $(c_n, \lambda_n, \nu_n)$, where $\nu_n = |\{m : m - \lambda_m \leqslant n < m\}|$ denotes the number of outstanding lags.

---

*Figure 5.* Online learning with drift penalization for linear losses.

To measure performance, we define the drift-penalized regret with weight $W \geqslant 0$ against a comparator $u \in \mathcal{K}$ as

$$\mathfrak{R}_N^{\text{drift}}(u; W) = \mathfrak{R}_N(u) + W \mathfrak{D}_N, \tag{3}$$

where $\mathfrak{R}_N(u) = \sum_{n=1}^{N}\langle c_n, z_n - u\rangle$ is the standard regret and $\mathfrak{D}_N = \sum_{n=1}^{N}\|z_{n-\lambda_n} - z_n\|$ is the total drift penalty. Any online linear optimization algorithm applies to this setting, with its standard regret bounding $\mathfrak{R}_N(u)$; more stable algorithms naturally incur smaller drift penalty $\mathfrak{D}_N$.

**P-FTRL and OMD Updates.** We analyze how Proximal Follow-The-Regularized-Leader (P-FTRL) and Online Mirror Descent (OMD) perform in online learning with drift penalization. Given a non-increasing learning rate sequence $\{\eta_n\}_{n=1}^{N} \subset (0, \infty)$ and initial point $z_1 \in \mathcal{K}$, these algorithms select predictions sequentially according to the following respective update rules:

$$\text{P-FTRL}: z_{n+1} = \operatorname*{argmin}_{z \in \mathcal{K}} \sum_{m=1}^{n} \left(\langle c_m, z\rangle + \frac{\alpha_n \|z_m - z\|^2}{2}\right), \quad (4)$$

$$\text{OMD}: z_{n+1} = \operatorname*{argmin}_{z \in \mathcal{K}} \left(\langle c_n, z\rangle + \frac{\|z_n - z\|^2}{2\eta_n}\right), \quad (5)$$

where $\{\alpha_n\}_{n=1}^{N}$ is given by $\alpha_1 = \frac{1}{\eta_1}, \alpha_{n+1} = \frac{1}{\eta_{n+1}} - \frac{1}{\eta_n}$. The following theorem provides regret and drift guarantees.

**Theorem 4.1.** *For every $u \in \mathcal{K}$, predictions $\{z_n\}_{n=1}^{T}$ generated by P-FTRL (4) or OMD (5) satisfy*

$$\mathfrak{R}_N(u) \leqslant \sum_{n=1}^{N} \frac{\frac{1}{\eta_n} - \frac{1}{\eta_{n-1}}}{2} \|z_n - u\|^2 + \frac{1}{2}\sum_{n=1}^{N} \eta_n \|c_n\|_\star^2,$$
$$\mathfrak{D}_N \leqslant \sum_{n=1}^{N} \eta_n \nu_n \|c_n\|_\star.$$

The regret bound is standard (e.g., Orabona (2025)). The drift bound arises by decomposing drift into single-step terms satisfying $\|z_{n+1} - z_n\| \leqslant \eta_n \|c_n\|_\star$; the resulting dependence on $\nu_n$ is the key to our delay-adaptive analysis. We also establish a novel drift guarantee for P-FTRL, which is essential for our delayed BCO results.

**Lemma 4.2.** *P-FTRL (4) generates $\{z_n\}_{n=1}^{T}$ such that*

$$\mathfrak{D}_N \leqslant \sum_{n=1}^{N} \eta_n \left\|\sum_{m=n-\lambda_n}^{n-1} c_m\right\|_\star + D\,H_\eta,$$

*where $H_\eta = \sum_{n=1}^{N}(1 - \frac{\eta_n}{\eta_{n-\lambda_n}})$ and $D$ is an upper bound on the diameter of $\mathcal{K}$ (Assumption 2.1).*

# 5. Blackbox Reductions and Main Results

Here, we develop algorithms for delayed OCO and BCO via reduction to drift-penalized online linear optimization. The key construction is a wrapper $\mathcal{W}_{\text{OCO}}$ (Algorithm 1) that takes any algorithm $\mathcal{B}$ for drift-penalized OLO as a blackbox and produces a new algorithm $\mathcal{W}_{\text{OCO}}(\mathcal{B})$ for delayed OCO. The wrapper feeds observation-ordered gradients $\widetilde{g}_n = \nabla \widetilde{f}_n(\widetilde{x}_n)$ as loss vectors to $\mathcal{B}$, with dual-delays $\widetilde{d}_n^\star$ as lags and dual-backlogs $\widetilde{\sigma}_n^\star$ as numbers of outstanding lags. The base algorithm $\mathcal{B}$ produces base-predictions that the wrapper outputs as its own predictions. An analogous

wrapper $\mathcal{W}_{\text{BCO}}$ (Algorithm 2) handles bandit feedback by using single-point gradient estimates from the classical work by Flaxman et al. (2004).

*Remark* 5.1 (Feasibility). This reduction is feasible because (i) the dual-backlog $\widetilde{\sigma}_n^\star$ equals the number of outstanding dual-delays by Lemma 3.6, and (ii) the wrapper runs updates on $\mathcal{B}$ only at observation times $\widetilde{r}_n$, when all required quantities are available. Specifically, at time $\widetilde{r}_n$, the gradient $\widetilde{g}_n = \nabla \widetilde{f}_n(\widetilde{x}_n)$ (or the value $\widetilde{f}_n(\widetilde{x}_n)$ in the case of bandit feedback) is revealed, while the dual-delay $\widetilde{d}_n^\star = |\{m : \widetilde{l}_n < \widetilde{r}_m < \widetilde{r}_n\}|$ and the dual-backlog $\widetilde{\sigma}_n^\star = |\{m : \widetilde{l}_m < \widetilde{r}_n < \widetilde{r}_m\}|$ can be computed from the observed history up to time $\widetilde{r}_n$.

## 5.1. Delayed OCO via Drift-Penalized OLO

We now present the wrapper $\mathcal{W}_{\text{OCO}}$ for delayed online convex optimization with first-order feedback. The wrapper maintains a current base-prediction $\bar{z}$ initialized by $\mathcal{B}$, outputs $\bar{z}$ whenever a prediction is requested, and updates $\bar{z}$ by forwarding each received gradient to $\mathcal{B}$ along with the dual-delay and dual-backlog.

---
**Algorithm 1** $\mathcal{W}_{\text{OCO}}(\mathcal{B})$

**Access :** Algorithm $\mathcal{B}$ for OLO with drift penalization.

Initialize $\bar{z}$ with initial output of $\mathcal{B}$.
**For** round $t = 1, 2, \ldots, T$:
1. Predict $x_t = \bar{z}$.
2. **For** each $s$ such that $s + d_s = t$, in observation order:
   • Receive $(s, \nabla f_s(x_s))$, send $(\nabla f_s(x_s), d_s^\star, \sigma_s^\star)$ to $\mathcal{B}$, and set $\bar{z}$ to the new output of $\mathcal{B}$.
---

By construction, this algorithm is steady: it updates only upon receiving feedback and outputs the current base-prediction otherwise. Effectively, the algorithm updates $\mathcal{B}$ sequentially on observation-ordered tuples $(\widetilde{g}_n, \widetilde{d}_n^\star, \widetilde{\sigma}_n^\star)$, where $\widetilde{g}_n = \nabla \widetilde{f}_n(\widetilde{x}_n)$ (i.e., $g_t = \nabla f_t(x_t)$), producing base-predictions $z_n$ at observation times $\widetilde{r}_{n-1}$ that yield predictions $\widetilde{x}_m = z_{m-\widetilde{d}_m^\star}$ at prediction times $\widetilde{l}_m \in (\widetilde{r}_{m-\widetilde{d}_m^\star - 1}, \widetilde{r}_{m-\widetilde{d}_m^\star})$.

The following theorem shows that the regret of $\mathcal{W}_{\text{OCO}}(\mathcal{B})$ is controlled through the drift-penalized regret of $\mathcal{B}$.

**Theorem 5.2** (Regret of Algorithm 1). *For any base algorithm $\mathcal{B}$ and comparator $u \in \mathcal{K}$, $\mathcal{W}_{\text{OCO}}(\mathcal{B})$ guarantees*

$$R_T(u) \leqslant \mathfrak{R}_T^{\text{drift}}(u; G),$$

*where $\mathfrak{R}_T^{\text{drift}}(u; W)$ is the drift-penalized regret (3) of $\mathcal{B}$ for loss vectors $c_n = \widetilde{g}_n$ and lags $\lambda_n = \widetilde{d}_n^\star$.*

*Under $\lambda$-strong convexity (2.3), it further holds that*

$$R_T(u) \leqslant \mathfrak{R}_T^{\text{drift}}(u; 3G) - \frac{\lambda}{2}\sum_{n=1}^{T}\|z_n - u\|^2.$$

Instantiating $\mathcal{B}$ with P-FTRL (4) or OMD (5) with appropriate learning rates yields the following guarantees.

**Corollary 5.3.** $\mathcal{W}_{\mathrm{OCO}}(\mathcal{B})$, where $\mathcal{B}$ runs P-FTRL (4) or OMD (5) with $\eta_n = \frac{D/G}{\sqrt{n+\sum_{m=1}^n \widetilde{\sigma}_m^\star}}$, guarantees

$$R_T(x^*) = O\left(GD\sqrt{T+d_{tot}}\right).$$

**Corollary 5.4.** Under $\lambda$-strong convexity (2.3), $\mathcal{W}_{\mathrm{OCO}}(\mathcal{B})$, where $\mathcal{B}$ runs P-FTRL (4) with $\eta_n = \frac{1}{n\lambda}$, guarantees

$$R_T(x^*) = O\left(\frac{G^2}{\lambda}\min\left\{(\sigma_{\max}\ln T,\ \sqrt{d_{tot}}\right\} + \ln T\right).$$

Running OMD (5) with $\eta_n = \frac{1}{n\lambda}$ instead guarantees

$$R_T(x^*) = O\left(\frac{G^2}{\lambda}(\sigma_{\max}+1)\ln T\right).$$

In both corollaries, the learning rates form a deterministic sequence that can be computed online: $\eta_n$ depends only on $n$ and the dual-backlogs $\widetilde{\sigma}_1^\star, \ldots, \widetilde{\sigma}_n^\star$, all of which are available at the $n$-th update. The proofs apply Theorem 4.1 and Lemma 4.2 with our chosen learning rates. Details are deferred to Appendix D.

### 5.2. Delayed BCO via Drift-Penalized OLO

We adapt the wrapper approach to bandit convex optimization, where only scalar loss values are observed, by using single-point gradient estimation from Flaxman et al. (2004).

Under Assumption 2.5, i.e., $r\mathbb{B}^k \subseteq \mathcal{K} \subseteq R\mathbb{B}^k$, we have $(1-\delta/r)\mathcal{K} + \delta\mathbb{B}^k \subseteq \mathcal{K}$ for all $\delta \in (0, r]$. This allows us to define, for any $\delta \in (0, r]$ and integrable $f : \mathcal{K} \to \mathbb{R}$, the $\delta$-smoothing of $f$ as $f^\delta : (1-\delta/r)\mathcal{K} \to \mathbb{R}$ given by $f^\delta(x) = \mathbb{E}_{v\sim\mathrm{Unif}(\mathbb{B}^k)}[f(x+\delta v)]$. The key properties of this construction are summarized in the following theorem.

**Theorem 5.5** (Single-Point Gradient Estimation, Flaxman et al. (2004)). *For any $\delta \in (0, r]$ and integrable $f : \mathcal{K} \to \mathbb{R}$:*

1. *The $\delta$-smoothing $f^\delta$ is differentiable with gradients $\nabla f^\delta(x) = \frac{k}{\delta}\mathbb{E}_{u\sim\mathrm{Unif}(\mathbb{S}^{k-1})}[f(x+\delta u)u]$.*
2. *If $f$ is convex ($\lambda$-strongly convex), so is $f^\delta$.*
3. *If $f$ is $G$-Lipschitz, then $|f^\delta(x) - f(x)| \leq G\delta$ and $\|\nabla f^\delta(x)\| \leq G$ for $x \in (1-\delta/r)\mathcal{K}$.*

The wrapper $\mathcal{W}_{\mathrm{BCO}}$ (Algorithm 2) follows the structure of $\mathcal{W}_{\mathrm{OCO}}$, but replaces true gradients with single-point estimates. The algorithm takes as input a non-increasing sequence of smoothing parameters $(\delta_t)_{t\geq 1} \subset (0, r]$, that can be specified independently of horizon $T$ when it's unknown. In round $t$, given the current base-prediction $y_t = \bar{z}$ from $\mathcal{B}$, the algorithm samples an exploration direction $u_t \sim \mathrm{Unif}(\mathbb{S}^{k-1})$, predicts $x_t = (1-\delta_t/r)y_t + \delta_t u_t$, and upon receiving loss $f_s(x_s)$ constructs the gradient estimate $\widehat{g}_s^\delta = \frac{k}{\delta_s}f_s(x_s)u_s$, which it forwards to $\mathcal{B}$ along with dual-delay $\widetilde{d}_s^\star$ and dual-backlog $\sigma_s^\star$ to update $\bar{z}$. By Theorem 5.5, we have $\mathbb{E}[\widehat{g}_t^\delta \mid y_t] = \nabla f_t^{\delta_t}((1-\delta_t/r)y_t)$ for all $t \in [T]$.

---

**Algorithm 2** $\mathcal{W}_{BCO}(\mathcal{B})$

---

**Access :** Algorithm $\mathcal{B}$ for OLO with drift penalization.
**Parameters :** Non-increasing sequence $(\delta_t)_{t\geq 1} \subset (0, r]$.
Initialize $\bar{z}$ with initial output of $\mathcal{B}$.
**For** round $t = 1, 2, \ldots, T$:
1. Sample $u_t \sim \mathrm{Unif}(\mathbb{S}^{k-1})$; predict $x_t = (1-\frac{\delta_t}{r})\bar{z} + \delta_t u_t$.
2. **For** each $s$ such that $s + d_s = t$, in observation order:
   - Receive $(s, f_s(x_s))$, send $(\frac{k}{\delta_s}f_s(x_s)u_s, d_s^\star, \sigma_s^\star)$ to $\mathcal{B}$, and set $\bar{z}$ to the new output of $\mathcal{B}$.

---

Unlike the OCO case, the resulting algorithm is not steady due to random exploration, but is steady in expectation for a fixed schedule $\delta_t = \delta$. Nevertheless, since the current base-prediction is updated only upon receiving feedback, the same analysis applies. The algorithm updates $\mathcal{B}$ sequentially on tuples $(\widetilde{\widehat{g}}_n^\delta, \widetilde{d}_n^\star, \widetilde{\sigma}_n^\star)$, where $\widetilde{\widehat{g}}_n^\delta = \frac{k}{\delta_n}\widetilde{f}_n(\widetilde{x}_n)\widetilde{u}_n$, producing base-predictions $z_n$ that yield predictions $\widetilde{x}_n = (1-\widetilde{\delta}_n/r)\widetilde{y}_n + \widetilde{\delta}_n\widetilde{u}_n$ with $\widetilde{y}_n = z_{n-\widetilde{d}_n^\star}$.

The following theorem bounds the expected regret of $\mathcal{W}_{\mathrm{BCO}}(\mathcal{B})$ in terms of the drift-penalized regret of $\mathcal{B}$, with an additional bias of order $\delta_{\mathrm{tot}} = \sum_{t=1}^T \delta_t$ due to smoothing.

**Theorem 5.6** (Regret of Algorithm 2). *For any base-algorithm $\mathcal{B}$, non-increasing sequence $\{\delta_t\}_{t\geq 1} \subset (0, r]$, and comparator $u \in \mathcal{K}$, $\mathcal{W}_{\mathrm{BCO}}(\mathcal{B})$ guarantees*

$$\bar{R}_T(u) \leq \mathbb{E}\left[\mathfrak{R}_T^{\mathrm{drift}}(u; G)\right] + \frac{6GR\delta_{\mathrm{tot}}}{r},$$

*where $\mathfrak{R}_T^{\mathrm{drift}}(u; W)$ is the drift-penalized regret (3) of $\mathcal{B}$ for loss vectors $c_n = \widetilde{\widehat{g}}_n^\delta$ and lags $\lambda_n = \widetilde{d}_n^\star$.*

*Under $\lambda$-strong convexity (2.3), it further holds that*

$$\bar{R}_T(u) \leq \mathbb{E}\left[\mathfrak{R}_T^{\mathrm{drift}}(u; 3G) - \frac{\lambda}{2}\sum_{n=1}^T \|z_n - u\|^2\right] + \frac{10GR\delta_{\mathrm{tot}}}{r}.$$

Selecting an appropriate smoothing sequence $(\delta_t)_{t=1}^T$ and instantiating $\mathcal{B}$ with P-FTRL (4) yields concrete guarantees. To state them, we introduce the *smoothness ratio* $\nu = \frac{M}{Gr}$, measuring gradient magnitude relative to the curvature-adjusted domain size, where $M$ is the absolute value bound (Assumption 2.4). Since the $n$-th update occurs in round $\rho(n) + d_{\rho(n)}$, the observation round of feedback from round $\rho(n)$, the smoothing parameter $\delta_n' = \delta_{\rho(n)+d_{\rho(n)}}$ is available and can be incorporated into the learning rate $\eta_n$.

**Corollary 5.7.** $\mathcal{W}_{\mathrm{BCO}}(\mathcal{B})$, where $(\delta_t)_{t\geq 1}$ is set to either a round-dependent schedule $\delta_t = r\min\left\{1, \frac{\sqrt{\nu k}}{t^{1/4}}\right\}$ or a fixed schedule $\delta_t = r\min\left\{1, \frac{\sqrt{\nu k}}{T^{1/4}}\right\}$ and $\mathcal{B}$ runs P-FTRL (4) with $\eta_n = \frac{D/G}{\sqrt{n+\sum_{m=1}^n (\widetilde{\sigma}_m^\star + (\nu k r/\delta_m')^2)}}$, guarantees

$$\bar{R}_T(x^*) = O\left(GD\left[\sqrt{d_{tot}} + T^{3/4}\sqrt{\nu k}\right]\right).$$

**Corollary 5.8.** *Under $\lambda$-strong convexity (2.3), $\mathcal{W}_{\mathrm{BCO}}(\mathcal{B})$, where $(\delta_t)_{t\geqslant 1}$ is set to either a round-dependent schedule $\delta_t = r\min\left\{1, (\frac{\nu^2 k^2 \log t}{t})^{1/3}\right\}$ or a fixed schedule $\delta_t = r\min\left\{1, (\frac{\nu^2 k^2 \log T}{T})^{1/3}\right\}$ and $\mathcal{B}$ runs P-FTRL (4) with $\eta_n = \frac{1}{n\lambda}$, guarantees*

$$\bar{R}_T(x^*) = O\Big(\frac{G^2}{\lambda}\Big[\min\{\sigma_{\max}\ln T, \sqrt{d_{tot}}\} + T^{2/3}(\ln T)^{1/3}(\nu k)^{2/3}\Big]\Big).$$

The proofs of both corollaries combine Theorem 4.1 and Lemma 4.2, using a concentration bound for $\|\sum_{m=n-\tilde{d}_n^\star}^{n-1} \tilde{\tilde{g}}_m^\delta\|$ to control the drift term in Appendix D.

## 6. Skipping Scheme

The $O(\sqrt{d_{\mathrm{tot}}})$ term in Corollaries 5.3 and 5.7 is pessimistic when delays are highly unbalanced: if a few rounds have exceptionally large delays, it is preferable to skip them (i.e., accept full loss and ignore their feedback), improving the bound to $O(\min_{Q\subseteq[T]}\{|Q| + \sqrt{\sum_{t\notin Q} d_t}\})$. For instance, following Thune et al. (2019), if the first $\lfloor\sqrt{T}\rfloor$ rounds have delay $\Theta(T)$ and the rest have 1, we get improvement from $O(T^{3/4})$ to $O(\sqrt{T})$. However, as delays are revealed upon feedback arrival, achieving this adaptively is nontrivial.

To address this, we adopt the skipping technique of Zimmert & Seldin (2020) as an external wrapper, $\mathcal{S}_{\mathrm{ZS}}$ (Algorithm 3), making it applicable to any delayed OCO algorithm $\mathcal{A}$. The wrapper acts as an interface between $\mathcal{A}$ and the environment, maintaining a tracking set $S$ of rounds awaiting feedback. When feedback for $s \in S$ arrives, it is forwarded to $\mathcal{A}$; if $s$ remains pending for too long, it is "skipped" by forwarding zero-valued feedback and removing $s$ from $S$. Thus, in effect, $\mathcal{A}$ is presented with a different delayed OCO problem, one with shorter delays and zero losses on skipped rounds.

---

**Algorithm 3** $\mathcal{S}_{\mathrm{ZS}}(\mathcal{A})$

**Access:** Algorithm $\mathcal{A}$ for OCO (BCO) with delays.

Initialize tracking set $S = \varnothing$ and set $\mathcal{D}_0 = 0$.
**For** round $t = 1, 2, \ldots, T$:
1. Set $\mathcal{D}_t = \mathcal{D}_{t-1} + |S|$.
2. **For** all $s \in S$ such that $t - s > \sqrt{\mathcal{D}_t}$:
   - Forward $(s, 0)$ to $\mathcal{A}$ and remove $s$ from $S$.
3. Query $\mathcal{A}$ for the next decision, obtain $x_t$, play $x_t$, and insert $t$ into the tracking set $S$.
4. **For** all $s$ such that $s + d_s = t$, in observation order:
   - Receive $(s, v_s)$ from the environment.
   - **If** $s \in S$ **then** forward $(s, v_s)$ to $\mathcal{A}$, remove $s$ from $S$.

---

By construction, for every $t \in [T]$, the wrapper passes feedback with index $t$ to $\mathcal{A}$ at round $t + d_t'$ for some $d_t' \in$

$\{0, \ldots, d_t\}$: either the true feedback $(t, v_t)$ upon arrival or dummy feedback $(t, 0)$ forwarded preemptively. Let $R^* \subseteq [T]$ denote the set of skipped rounds, i.e., those for which dummy feedback was forwarded, and define $f_t' = f_t \, \mathbb{I}(t \notin R^*)$. Then, in effect, $\mathcal{S}_{\mathrm{ZS}}$ presents $\mathcal{A}$ with a delayed OCO problem having losses $\{f_t'\}_{t=1}^T$ and delays $\{d_t'\}_{t=1}^T$. The following theorem bounds the regret of $\mathcal{S}_{\mathrm{ZS}}(\mathcal{A})$.

**Theorem 6.1.** *The regret $R_T(u)$ of $\mathcal{S}_{\mathrm{ZS}}(\mathcal{A})$ against any comparator $u \in \mathcal{K}$ satisfies $R_T(u) \leqslant R_T'(u) + GD|R^*|$, where $R_T'(u) = \sum_{t=1}^T (f_t'(x_t) - f_t'(u))$ is the regret of $\mathcal{A}$ on delayed OCO problem with losses and delays $\{f_t', d_t'\}_{t=1}^T$. Moreover, the number of skipped rounds $|R^*|$ and the total modified delay $d_{tot}' = \sum_{t=1}^T d_t'$ satisfy*

$$|R^*| + \sqrt{d_{tot}'} = O\left(\min_{Q\subseteq[T]}\left\{|Q| + \sqrt{\sum_{t\notin Q} d_t}\right\}\right).$$

Composing $\mathcal{S}_{\mathrm{ZS}}$ with wrappers $\mathcal{W}_{\mathrm{OCO}}$ and $\mathcal{W}_{\mathrm{BCO}}$ immediately improves upon Corollaries 5.3 and 5.7, respectively.

**Corollary 6.2.** *Under the conditions of Corollaries 5.3 and 5.7, applying the skipping wrapper yields:*

$$R_T(x^*) = O\left(GD\Big[\min_{Q\subseteq[T]}\Big\{|Q| + \sqrt{\sum_{t\notin Q} d_t}\Big\} + \sqrt{T}\Big]\right),$$
$$\bar{R}_T(x^*) = O\left(GD\Big[\min_{Q\subseteq[T]}\Big\{|Q| + \sqrt{\sum_{t\notin Q} d_t}\Big\} + T^{\frac{3}{4}}\sqrt{\nu k}\Big]\right),$$

*for $\mathcal{S}_{\mathrm{ZS}}(\mathcal{W}_{\mathrm{OCO}}(\mathcal{B}))$ and $\mathcal{S}_{\mathrm{ZS}}(\mathcal{W}_{\mathrm{BCO}}(\mathcal{B}))$, respectively.*

The proofs are deferred to Appendix E.

## 7. Discussion and Future Work

We introduced a continuous-time model for online learning with delayed feedback that places prediction and observation events on a single timeline. This observation-centric view yields a blackbox reduction to drift-penalized online linear optimization: wrapping any base algorithm with regret and stability control produces a delayed learner whose regret is governed by the base algorithm's drift-penalized regret under the induced observation ordering.

Several directions remain open. First, our analysis assumes an oblivious adversary that fixes losses and delays in advance. Extending the framework to adaptive adversaries, where delays may depend on the learner's actions, is both technically challenging and relevant to congestion control and strategic environments. Second, delayed feedback is ubiquitous in distributed optimization with asynchronous communication. Our model may provide a useful abstraction for multi-agent online learning with heterogeneous, time-varying delays, including federated learning systems.

## Impact Statement

This paper presents work whose goal is to advance the field of Machine Learning. There are many potential societal consequences of our work, none which we feel must be specifically highlighted here.

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

## A. General facts

*Fact* A.1. Under Assumptions 2.2 and 2.3, the diameter of $\mathcal{K}$ is bounded by $\frac{2G}{\lambda}$, i.e., $\sup_{x,y\in\mathcal{K}}\|x-y\| \leqslant \frac{2G}{\lambda}$.

*Proof.* For all $x, y \in \mathcal{K}$, we have $\lambda\|x-y\|^2 \leqslant \langle \nabla f(x) - \nabla f(y),\, x - y \rangle \leqslant 2G\|x-y\|$.

Hence, $D \leqslant \frac{2G}{\lambda} < \infty$. $\qquad\square$

*Fact* A.2 (Rearrangement Inequality). For every choice of $N \in \mathbb{N}$, real numbers $x_1 \leqslant x_2 \leqslant ... \leqslant x_N$, $y_1 \leqslant y_2 \leqslant ... \leqslant y_N$, and index permutation $\rho : [N] \to [N]$, it holds that

$$x_1 y_N + ... + x_N y_1 \leqslant x_1 y_{\rho(1)} + ... x_N y_{\rho(N)} \leqslant x_1 y_1 + ... + x_N y_N.$$

*Proof.* This fact restates Theorem 368 in (Hardy et al., 1952). $\qquad\square$

*Fact* A.3 (Logarithmic Telescoping Inequality). Let $(X_n)_{n\geqslant 1}$ be non-negative reals such that $X_1 > 0$. Set $S_n = \sum_{m=1}^n X_m$. Then, for every $N \geqslant 1$, we have

$$\sum_{n=1}^N \frac{X_n}{S_n} \leqslant \log(eS_N/X_1).$$

*Proof.* Since $X_1 > 0$, all $S_n > 0$. For $n \geqslant 2$, since $S_n = S_{n-1} + X_n$,

$$\log S_n - \log S_{n-1} = \log\left(1 + \frac{X_n}{S_{n-1}}\right) \geqslant \frac{\frac{X_n}{S_{n-1}}}{1 + \frac{X_n}{S_{n-1}}} = \frac{X_n}{S_n},$$

using $\log(1 + u) \geqslant \frac{u}{1+u}$ for $u \geqslant 0$. Summing from $n = 1$ to $N$ gives

$$\sum_{n=1}^N \frac{X_n}{S_n} \leqslant 1 + \sum_{n=2}^N (\log S_n - \log S_{n-1}) = 1 + \log S_N - \log S_1 = \log(eS_N/X_1).$$

This concludes the proof. $\qquad\square$

*Fact* A.4 (Square-Root Telescoping Inequality). Let $(X_n)_{n\geqslant 1}$ be non-negative reals and $X_1 > 0$. Set $S_n = \sum_{m=1}^n X_m$. Then, for every $N \geqslant 1$, we have

$$\sum_{n=1}^N \frac{X_n}{\sqrt{S_n}} \leqslant 2\sqrt{S_N} - \sqrt{S_1}.$$

*Proof.* Since $X_1 > 0$, all $S_n > 0$. For $n \geqslant 2$, since $S_n = S_{n-1} + X_n$, it holds that

$$\sqrt{S_n} - \sqrt{S_{n-1}} = \frac{S_n - S_{n-1}}{\sqrt{S_n} + \sqrt{S_{n-1}}} = \frac{X_n}{\sqrt{S_n} + \sqrt{S_{n-1}}} \geqslant \frac{X_n}{2\sqrt{S_n}}.$$

Summing over $n = 1, \dots, N$, multiplying by 2, and telescoping,

$$\sum_{n=1}^N \frac{X_n}{\sqrt{S_n}} \leqslant \sqrt{S_1} + 2\sum_{n=2}^N \left(\sqrt{S_n} - \sqrt{S_{n-1}}\right) = 2\sqrt{S_N} - \sqrt{S_1}.$$

This concludes the proof. $\qquad\square$

## B. Technical Results for the Continuous Time Model: Proofs

This section establishes the technical results concerning the CTM (Definition 3.1). Recall the observation-ordering convention (Definition 3.5): $\tilde{a}_n = a_{\rho(n)}$, where $\rho$ is the permutation of $[T]$ satisfying $r_{\rho(1)} \leqslant \cdots \leqslant r_{\rho(T)}$. Fact B.1 serves as a central tool throughout this section, connecting delays and backlogs with their duals. Facts B.2 and B.3 are used in the proof of Corollary 5.7, while Fact B.4 plays a key role in the analysis of delay-adaptive learning rates.

*Fact* B.1. Define $\beta(t) = |\{s : r_s \leqslant r_t\}|$. For all $t \in [T]$, $\sigma_t^\star = d_t + t - \beta(t)$ and $d_t^\star = \sigma_t + \beta(t) - t$. Furthermore, $\beta = \rho^{-1}$ as permutations of $[T]$, and for all $n \in [T]$, $\tilde{\sigma}_n^\star = \tilde{d}_n + \rho(n) - n$ and $\tilde{d}_n^\star = \tilde{\sigma}_n + n - \rho(n)$.

*Proof.* From the definition of delay and backlog (1) as induced by the CTM, we have

$$t + d_t = |\{s : l_s < r_t\}| = |\{s : r_s \leqslant r_t\}| + |\{s : l_s < r_t < r_s\}| = \beta(t) + \sigma_t^\star,$$
$$t - \sigma_t = |\{s : r_s < l_t\} \cup \{t\}| = |\{s : r_s \leqslant r_t\}| - |\{s : l_t < r_s < r_t\}| = \beta(t) - d_t^\star.$$

Note that $\beta = \rho^{-1}$ as permutations of $[T]$, since $\beta(\rho(n)) = |\{s : r_s \leqslant r_{\rho(n)}\}| = n$ for all $n \in [T]$.

The remaining identities follow by substituting $t = \rho(n)$. $\qquad\square$

**Theorem 3.4** (Restated). *For quantities in (1) and (2), it holds that:*

(a) $\sum_{t=1}^T d_t = \sum_{t=1}^T \sigma_t = \sum_{t=1}^T d_t^\star = \sum_{t=1}^T \sigma_t^\star$,
(b) $\sigma_{\max} = \sigma_{\max}^\star$, $\frac{1}{2} d_{\max} \leqslant d_{\max}^\star \leqslant 2 d_{\max}$,
(c) $d_t^\star = \sigma_t + \beta(t) - t$, $\sigma_t^\star = d_t + t - \beta(t)$,

*where $\beta : [T] \to [T]$ is the permutation $\beta(t) = |\{s : r_s \leqslant r_t\}|$.*

*Proof.* We address each claim separately.

(a) To show the classical result $\sum_{t=1}^T d_t = \sum_{t=1}^T \sigma_t$ simply swap the order of summation:

$$\sum_{t=1}^T d_t = \sum_{t=1}^T \sum_{s=1}^T \mathbb{I}(l_s \in I_t) = \sum_{s=1}^T \sum_{t=1}^T \mathbb{I}(l_s \in I_t) = \sum_{s=1}^T \sigma_s.$$

The identities $\sum_{t=1}^T \sigma_t^\star = \sum_{t=1}^T d_t$ and $\sum_{t=1}^T d_t^\star = \sum_{t=1}^T \sigma_t$ follow immediately from Fact B.1, because $\beta$ is a permutation of $[T]$ and $\sum_{t=1}^T \beta(t) = \sum_{t=1}^T t$.

(b) To show that $\sigma_{\max}^\star \leqslant \sigma_{\max}$, note that $r_t \in (l_{t+d_t}, l_{t+d_t+1})$ by Lemma 3.2, and so

$$\{s : r_t \in I_s\} \subseteq (\{t + d_t\} \cup \{s : l_{t+d_t} \in I_s\}) \backslash \{t\}.$$

Hence, $\sigma_t^\star \leqslant \sigma_{t+d_t} + 1 - 1 = \sigma_{t+d_t}$ for every $t \in [T]$, which means $\sigma_{\max}^\star \leqslant \sigma_{\max}$.

To show that $\sigma_{\max}^\star \geqslant \sigma_{\max}$, note that $\tilde{l}_n \in (\tilde{r}_{n-\tilde{d}_n^\star-1}, \tilde{r}_{n-\tilde{d}_n^\star})$ by Lemma 3.6, and so

$$\{m : \tilde{l}_n \in \tilde{I}_m\} \subseteq (\{n - \tilde{d}_n^\star\} \cup \{m : \tilde{r}_{n-\tilde{d}_n^\star} \in \tilde{I}_m\}) \backslash \{n\}.$$

Hence, $\tilde{\sigma}_n \leqslant \tilde{\sigma}_{n-\tilde{d}_n^\star}^\star + 1 - 1 = \tilde{\sigma}_{n-\tilde{d}_n^\star}^\star$ for every $n \in [T]$, which means $\sigma_{\max} \leqslant \sigma_{\max}^\star$.

To show that $d_{\max}^\star \leqslant 2 d_{\max}$, note that for each $t \in [T]$, $\{s : r_s \in I_t\} \subseteq \{s : l_s \in I_t\} \cup \{s : l_t \in I_s\}$. Hence, $d_{\max}^\star \leqslant \max_t(\sigma_t + d_t) \leqslant \sigma_{\max} + d_{\max} \leqslant 2 d_{\max}$, where the result $\sigma_{\max} \leqslant d_{\max}$ can be inferred from Lemma 3.2.

To show that $d_{\max} \leqslant 2 d_{\max}^\star$, note that for each $t \in [T]$, $\{s : l_s \in I_t\} \subseteq \{s : r_s \in I_t\} \cup \{s : r_t \in I_s\}$. Hence, $d_{\max} \leqslant \max_t(d_t^\star + \sigma_t^\star) \leqslant d_{\max}^\star + \sigma_{\max}^\star \leqslant 2 d_{\max}^\star$, where the result $\sigma_{\max}^\star \leqslant d_{\max}^\star$ can be inferred from Lemma 3.6.

(c) These identities are established in Fact B.1. $\qquad\square$

*Fact* B.2. For every $N \in [T]$, it holds that

$$\sum_{n=1}^N \tilde{d}_n^\star \leqslant \sum_{n=1}^N \tilde{\sigma}_n^\star.$$

*Proof.* For every $N \in [T]$, we can write

$$
\begin{aligned}
\sum_{n=1}^{N}(\tilde{\sigma}_n^\star - \tilde{d}_n^\star) &= \sum_{n=1}^{N}\sum_{m=1}^{T}(\mathbb{I}(\tilde{r}_n \in \tilde{I}_m) - \mathbb{I}(\tilde{r}_m \in \tilde{I}_n)) \\
&= \sum_{n=1}^{N}\sum_{m=1}^{N}(\mathbb{I}(\tilde{r}_n \in \tilde{I}_m) - \mathbb{I}(\tilde{r}_m \in \tilde{I}_n)) \\
&\quad + \sum_{n=1}^{N}\sum_{m=N+1}^{T}(\mathbb{I}(\tilde{r}_n \in \tilde{I}_m) - \mathbb{I}(\tilde{r}_m \in \tilde{I}_n)) \\
&= 0 + \sum_{n=1}^{N}\sum_{m=N+1}^{T}\mathbb{I}(\tilde{r}_n \in \tilde{I}_m),
\end{aligned}
$$

where the final equality used symmetry and the fact that $\tilde{r}_m \notin \tilde{I}_n$ for $m > N \geqslant n$. $\square$

*Fact* B.3. For every $t \in [T-1]$ and $n \in [T-1]$, it holds that

$$
\sigma_{t+1} \leqslant \sigma_t + 1, \quad \tilde{\sigma}_{n+1}^\star \geqslant \tilde{\sigma}_n^\star - 1.
$$

*Proof.* Using the definition of backlog and dual-backlog, we write

$$
\sigma_{t+1} = |\{s : l_{t+1} \in I_s\}| = |\{s \leqslant t : r_s > l_{t+1}\}| \leqslant |\{s \leqslant t : r_s > l_t\}| = \sigma_t + 1,
$$
$$
\tilde{\sigma}_n^\star = |\{m : \tilde{r}_n \in \tilde{I}_m\}| = |\{m > n : \tilde{l}_m < \tilde{r}_n\}| \leqslant |\{m > n : \tilde{l}_m < \tilde{r}_{n+1}\}| = \tilde{\sigma}_{n+1}^\star + 1. \quad \square
$$

*Fact* B.4. The following two inequalities hold

$$
\sum_{n=1}^{T} \frac{\tilde{\sigma}_n^\star}{n} \leqslant \sigma_{\max} \log(eT), \quad \sum_{n=1}^{T} \frac{\tilde{d}_n^\star}{n} \leqslant \min\left\{\sigma_{\max} \log(eT), 2\sqrt{d_{\text{tot}}}\right\}.
$$

*Proof.* To begin, observe the following: $\sigma_{\max}^\star = \sigma_{\max}$ by Theorem 3.4, $\sum_{n=1}^{T} \frac{1}{n} \leqslant \log(eT)$ by Fact A.3, and $\sum_{n=1}^{T} \frac{\rho(n)}{n} \geqslant T$ by Fact A.2. We use these observations freely throughout the following proof.

To prove the first inequality, simply write

$$
\sum_{n=1}^{T} \frac{\tilde{\sigma}_n^\star}{n} \leqslant \max_{n \in [T]} \tilde{\sigma}_n^\star \cdot \sum_{n=1}^{T} \frac{1}{n} \leqslant \sigma_{\max} \log(eT).
$$

We split the proof of the second inequality into two parts.

- To show that $\sum_{n=1}^{T} \frac{\tilde{d}_n^\star}{n} \leqslant \sigma_{\max} \log(eT)$, use Fact B.1 to write

$$
\sum_{n=1}^{T} \frac{\tilde{d}_n^\star}{n} = \sum_{n=1}^{T} \frac{\tilde{\sigma}_n + n - \rho(n)}{n} \leqslant \sigma_{\max} \log(eT) + T - \sum_{n=1}^{T} \frac{\rho(n)}{n} \leqslant \sigma_{\max} \log(eT).
$$

- To show that $\sum_{n=1}^{T} \frac{\tilde{d}_n^\star}{n} \leqslant 2\sqrt{d_{\text{tot}}}$, observe that for all $n \in [T]$ it holds that $n \geqslant \tilde{d}_n^\star + 1$ as the total number of observations up to and including time $\tilde{r}_n$ is equal to $n$. Hence, for all $n \in [T]$,

$$
n^2 \geqslant \sum_{m=1}^{n} m \geqslant \sum_{m=1}^{n}(\tilde{d}_m^\star + 1) \geqslant 1 + \sum_{m=1}^{n}\tilde{d}_m^\star.
$$

By Fact A.4, we have

$$
\sum_{n=1}^{T} \frac{\tilde{d}_n^\star}{n} \leqslant \left(1 + \sum_{n=1}^{T} \frac{\tilde{d}_n^\star}{\sqrt{1 + \sum_{m=1}^{n} \tilde{d}_m^\star}}\right) - 1 \leqslant 2\sqrt{1 + \sum_{m=1}^{T} \tilde{d}_m^\star} - 2 \leqslant 2\sqrt{d_{\text{tot}}}.
$$

This concludes the proof of both inequalities. $\square$

## C. Online Linear Optimization with Drift Penalization: Proofs

This section proves the results for drift-penalized OLO from Section 4. Theorem 4.1 and Lemma 4.2 establish how P-FTRL and OMD control the standard regret $\mathfrak{R}_N(u) = \sum_{n=1}^N \langle c_n, z_n - u \rangle$ and the prediction drift $\mathfrak{D}_N = \sum_{n=1}^N \|z_{n-\lambda_n} - z_n\|$. Given a non-increasing learning rate sequence $\{\eta_n\}_{n=1}^N \subset (0, \infty)$ and initial point $z_1 \in \mathcal{K}$, we consider the update rules

$$\text{P-FTRL}: \ z_{n+1} = \operatorname{argmin}_{z \in \mathcal{K}} \left\langle \sum_{m=1}^n c_m, z \right\rangle + \sum_{m=1}^n \frac{1/\eta_m - 1/\eta_{m-1}}{2} \|z_m - z\|^2,$$

$$\text{OMD}: \ z_{n+1} = \operatorname{argmin}_{z \in \mathcal{K}} \langle c_n, z \rangle + \frac{1/\eta_n}{2} \|z_n - z\|^2,$$

with the convention $1/\eta_0 = 0$.

### C.1. General results for FTRL and OMD

The following results are standard in the analysis of FTRL and OMD.

**Theorem C.1** (Orabona (2025), adapted from Lemmas 7.1 and 7.8 for linear losses). *Let $\mathcal{K} \subset \mathbb{R}^k$ be a convex compact set, let $\{\eta_n\}_{n=1}^N \subset (0, \infty)$ be a non-increasing sequence, and let $\psi_1, \ldots, \psi_N : \mathbb{R}^k \to \mathbb{R}$ be differentiable functions such that each $\psi_n$ is $(1/\eta_n)$-strongly convex with respect to $\|\cdot\|$ on $\mathcal{K}$. Given loss vectors $\{c_n\}_{n=1}^N \subset \mathbb{R}^k$ and an initial point $z_1 \in \mathcal{K}$, we can iteratively select unique minimizers $z_{n+1} = \operatorname{argmin}_{z \in \mathcal{K}} \{\psi_n(z) + \sum_{m=1}^n \langle c_m, z \rangle\}$. For proximal regularizers $\psi_n$ such that $z_n \in \operatorname{argmin}_{z \in \mathcal{K}} \{\psi_n(z) - \psi_{n-1}(z)\}$, for all $u \in \mathcal{K}$, it holds that*

$$\sum_{n=1}^N \langle c_n, z_n - u \rangle \leq \psi_N(u) + \sum_{n=1}^N \left( \tfrac{1}{2} \eta_n \|c_n\|_\star^2 + \psi_{n-1}(z_n) - \psi_n(z_n) \right).$$

**Lemma C.2** (Stability lemma, modified version of Lemma A.2 in Qiu et al. (2025)). *Let $\mathcal{K} \subset \mathbb{R}^k$ be a convex compact set. For $i \in \{0, 1\}$, let function $\phi_i : \mathbb{R}^k \to \mathbb{R}$ be differentiable and $\lambda_i$-strongly convex with respect to $\|.\|$ on $\mathcal{K}$. Consider $z_i \in \operatorname{argmin}_{x \in \mathcal{K}} \langle w_i, x \rangle + \phi_i(x)$. It holds that*

$$\tfrac{\lambda_0 + \lambda_1}{2} \|z_0 - z_1\|^2 \leq \langle w_0 - w_1, z_1 - z_0 \rangle + (\phi_1(z_0) - \phi_0(z_0)) - (\phi_1(z_1) - \phi_0(z_1)).$$

*Proof.* For $i \in \{0, 1\}$, let $h_i(x) = \langle w_i, x \rangle + \phi_i(x)$ denote the $\lambda_i$-strongly convex function minimized at point $z_i$. Therefore, by the strong convexity of $h_i$, it holds that

$$h_i(z_{1-i}) - h_i(z_i) \geq \tfrac{\lambda_i}{2} \|z_{1-i} - z_i\|^2.$$

By summing the above inequality for both $i \in \{0, 1\}$, we conclude

$$\langle w_0 - w_1, z_1 - z_0 \rangle + (\phi_1(z_0) - \phi_0(z_0)) - (\phi_1(z_1) - \phi_0(z_1)) = h_0(z_1) - h_0(z_0) + h_1(z_0) - h_1(z_1)$$
$$\geq \tfrac{\lambda_0 + \lambda_1}{2} \|z_0 - z_1\|^2. \qquad \square$$

**Lemma C.3** (Orabona (2025), adapted from Lemma 6.10 for linear losses). *Let $\mathcal{K} \subset \mathbb{R}^k$ be a convex compact set. Consider function $\psi : \mathbb{R}^k \to \mathbb{R}$ that is proper, closed, differentiable, 1-strongly convex function with respect to $\|.\|$ in $\mathcal{K}$, with $B_\psi : \mathbb{R}^k \times \mathbb{R}^k \to \mathbb{R}$ denoting its Bregman divergence, i.e., $B_\psi(x; y) = \psi(x) - \psi(y) - \langle \nabla \psi(y), x - y \rangle$. Given $\{c_n\}_{n=1}^N \subset \mathbb{R}^k$, non-increasing sequence $\{\eta_n\}_{n=1}^N \subset (0, \infty)$, and $z_1 \in \mathcal{K}$, we can iteratively select unique minimizers $z_{n+1} = \operatorname{argmin}_{z \in \mathcal{K}} \langle z, c_n \rangle + \frac{1}{\eta_n} B_\psi(z; z_n)$. Moreover, for all $u \in \mathcal{K}$ and $n \in [N]$:*

$$\eta_n \langle c_n, x_n - u \rangle \leq B_\psi(u; z_n) - B_\psi(u; z_{n+1}) + \tfrac{\eta_n^2}{2} \|c_n\|_\star^2.$$

**Theorem C.4** (OMD: Regret and Drift). *Under the assumptions of Lemma C.3, for all $u \in \mathcal{K}$, it holds that*

$$\sum_{n=1}^N \langle c_n, z_n - u \rangle \leq \sum_{n=1}^N \left( \tfrac{1}{\eta_n} - \tfrac{1}{\eta_{n-1}} \right) B_\psi(u; z_n) + \tfrac{1}{2} \sum_{n=1}^N \eta_n \|c_n\|_\star^2.$$

*Moreover, for all $n \in [N-1]$, it holds that $\|z_{n+1} - z_n\| \leq \eta_n \|c_n\|_\star$.*

*Proof.* For all $u \in \mathcal{K}$, using Lemma C.3, we write

$$\sum_{n=1}^N \langle c_n, z_n - u \rangle \leq \sum_{n=1}^N \left( \tfrac{1}{\eta_n} B_\psi(u; z_n) - \tfrac{1}{\eta_n} B_\psi(u; z_{n+1}) \right) + \tfrac{1}{2} \sum_{n=1}^N \eta_n \|c_n\|_\star^2$$
$$= \sum_{n=1}^N \left( \tfrac{1}{\eta_n} - \tfrac{1}{\eta_{n-1}} \right) B_\psi(u; z_n) - \tfrac{1}{\eta_N} B_\psi(u; z_{N+1}) + \tfrac{1}{2} \sum_{n=1}^N \eta_n \|c_n\|_\star^2,$$

where $B_\psi(u; z_{N+1}) \geq 0$ for Bregman divergence. This concludes the proof of the regret bound.

Fix arbitrary $n \in [N]$ and let $\phi_0(z) = \frac{1}{\eta_n} B_\psi(z; z_n)$. Let $g_0 = \nabla\phi_0(z_n)$ and $g_1 = \nabla\phi_0(z_{n+1})$. By the first-order optimality of $z_n = \operatorname{argmin}_{z\in\mathcal{K}}\{\phi_0(z)\}$ and $z_{n+1} = \operatorname{argmin}_{z\in\mathcal{K}}\{\langle z, c_n\rangle + \phi_0(z)\}$, we have

$$\langle g_0, z_n - z_{n+1}\rangle \leq 0 \quad \text{and} \quad \langle c_n + g_1, z_n - z_{n+1}\rangle \geq 0.$$

The monotonicity inequality for $\frac{1}{\eta_n}$-strong convex $\phi_0$ yields $\langle g_0 - g_1, z_n - z_{n+1}\rangle \geq \frac{1}{\eta_n}\|z_n - z_{n+1}\|^2$. Combining these inequalities, we have

$$
\begin{aligned}
\frac{1}{\eta_n}\|z_n - z_{n+1}\|^2 &\leq \langle g_0 - g_1, z_n - z_{n+1}\rangle \\
&= \langle c_n, z_n - z_{n+1}\rangle + \langle g_0, z_n - z_{n+1}\rangle - \langle c_n + g_1, z_n - z_{n+1}\rangle \\
&\leq \|c_n\|_\star\|z_n - z_{n+1}\| + 0 + 0.
\end{aligned}
$$

Consequently, $\|z_n - z_{n+1}\| \leq \eta_n\|c_n\|_\star$. $\qquad\square$

### C.2. Auxiliary Results for Drift Control of P-FTRL

**Lemma C.5.** *Let $\{z_n\}_{n=1}^T$ denote predictions of P-FTRL (4). For all $n \in [N]$ and $\lambda \in [N - n]$, it holds that*

$$\|z_{n+\lambda} - z_n\| \leq \eta_{n+\lambda}\|\textstyle\sum_{m=n}^{n+\lambda-1} c_m\|_\star + (1 - \eta_{n+\lambda}/\eta_n)D.$$

*Moreover, it holds that $\|z_{n+1} - z_n\| \leq \eta_n\|c_n\|_\star$ for $n \in [N - 1]$.*

*Proof.* Let $\alpha_1 = 1/\eta_1$ and $\alpha_{n+1} = 1/\eta_{n+1} - 1/\eta_n$ for $n \in [N - 1]$.

To apply Lemma C.2, consider the following vectors and functions

$$
\begin{aligned}
w_0 &= \textstyle\sum_{m=1}^{n-1} c_m, \quad \phi_0(z) = \textstyle\sum_{m=1}^n \frac{\alpha_m}{2}\|z - z_m\|^2, \\
w_1 &= \textstyle\sum_{m=1}^{n+\lambda-1} c_m, \quad \phi_1(z) = \textstyle\sum_{m=1}^{n+\lambda} \frac{\alpha_m}{2}\|z - z_m\|^2,
\end{aligned}
$$

so that $z_n = \operatorname{argmin}_{z\in\mathcal{K}}\langle z, w_0\rangle + \phi_0(z)$ and $z_{n+\lambda} = \operatorname{argmin}_{z\in\mathcal{K}}\langle z, w_1\rangle + \phi_1(z)$, while $\phi_0$ and $\phi_1$ are $1/\eta_n$ and $1/\eta_{n+\lambda}$-strongly convex, respectively.

Applying Lemma C.2, we have

$$
\begin{aligned}
\frac{\frac{1}{\eta_{n+\lambda}} + \frac{1}{\eta_n}}{2}\|z_n - z_{n+\lambda}\|^2 &\leq \langle\textstyle\sum_{m=n}^{n+\lambda-1} c_m, z_n - z_{n+\lambda}\rangle + \textstyle\sum_{m=n+1}^{n+\lambda} \frac{\alpha_m}{2}\left(\|z_n - z_m\|^2 - \|z_{n+\lambda} - z_m\|^2\right) \\
&= \langle\textstyle\sum_{m=n}^{n+\lambda-1} c_m, z_n - z_{n+\lambda}\rangle + \langle\textstyle\sum_{m=n+1}^{n+\lambda} \frac{\alpha_m}{2}(z_n + z_{n+\lambda} - 2z_m), z_n - z_{n+\lambda}\rangle \\
&= \langle\textstyle\sum_{m=n}^{n+\lambda-1} c_m, z_n - z_{n+\lambda}\rangle + \langle\textstyle\sum_{m=n+1}^{n+\lambda} \alpha_m(z_n - z_m), z_n - z_{n+\lambda}\rangle \\
&\quad - \frac{\frac{1}{\eta_{n+\lambda}} - \frac{1}{\eta_n}}{2}\|z_n - z_{n+\lambda}\|^2.
\end{aligned}
$$

Grouping terms containing $\|z_n - z_{n+\lambda}\|^2$ on the left side and using Assumptions 2.1, we write

$$
\begin{aligned}
\frac{1}{\eta_{n+\lambda}}\|z_n - z_{n+\lambda}\|^2 &\leq \langle\textstyle\sum_{m=n}^{n+\lambda-1} c_m, z_n - z_{n+\lambda}\rangle + \langle\textstyle\sum_{m=n+1}^{n+\lambda} \alpha_m(z_n - z_m), z_n - z_{n+\lambda}\rangle \\
&\leq \left(\|\textstyle\sum_{m=n}^{n+\lambda-1} c_m\|_\star + (1/\eta_{n+\lambda} - 1/\eta_n)D\right)\|z_n - z_{n+\lambda}\|.
\end{aligned}
$$

The first inequality immediately follows. Alternatively, for $\lambda = 1$, we have

$$
\begin{aligned}
\frac{1}{\eta_n}\|z_n - z_{n+1}\|^2 &= \left(\frac{1}{\eta_{n+1}} - \alpha_{n+1}\right)\|z_n - z_{n+1}\|^2 \\
&\leq \langle c_n, z_n - z_{n+1}\rangle + \langle\alpha_{n+1}(z_n - z_{n+1}), z_n - z_{n+1}\rangle - \alpha_{n+1}\|z_n - z_{n+1}\|^2 \\
&= \langle c_n, z_n - z_{n+1}\rangle \\
&\leq \|c_n\|_\star\|z_n - z_{n+1}\|.
\end{aligned}
$$

The second result follows from this, concluding the proof. $\qquad\square$

**Lemma C.6.** *For learning rates $\{\eta_n\}_{n=1}^N$ of P-FTRL* (4)*, let $H_\eta = \sum_{n=1}^N (1 - \frac{\eta_n}{\eta_{n-\lambda_n}})$. Then,*

$$H_\eta \leqslant (\max_{n\in[N]} \nu_n + 1) \log\left(\frac{e\eta_1}{\eta_N}\right) \qquad and \qquad H_\eta \leqslant \frac{\max_{n\in[N]} \sum_{m=n}^{n+\nu_n} \eta_m}{\eta_N}.$$

*Proof.* Let $\alpha_1 = 1/\eta_1 > 0$ and $\alpha_{n+1} = 1/\eta_{n+1} - 1/\eta_n \geqslant 0$ for $n \in [N-1]$. Then, we can write

$$H_\eta = \sum_{n=1}^N \eta_n \left(\frac{1}{\eta_n} - \frac{1}{\eta_{n-\lambda_n}}\right) = \sum_{n=1}^N \eta_n \sum_{m=1}^N \alpha_m \, \mathbb{I}(n - \lambda_n < m \leqslant n)$$

$$= \sum_{m=1}^N \alpha_m \sum_{n=1}^N \eta_n \, \mathbb{I}(n - \lambda_n < m \leqslant n).$$

Since the sequence $\{\eta_n\}_{n=1}^N$ is non-increasing and by Lemma 3.6,

$$\sum_{n=1}^N \mathbb{I}(n - \lambda_n < m \leqslant n) \leqslant \sum_{n=1}^N \mathbb{I}(n - \lambda_n \leqslant m < n) + 1 = \nu_m + 1,$$

we can further write, with $\nu_{\max} = \max_{n\in[N]} \nu_n$:

$$H_\eta \leqslant \sum_{m=1}^N \alpha_m \eta_m \left(\nu_m + 1\right) \leqslant (\nu_{\max} + 1) \sum_{m=1}^N \alpha_m \eta_m$$

$$= (\nu_{\max} + 1) \sum_{m=1}^N \frac{\alpha_m}{\sum_{k=1}^m \alpha_k} \leqslant (\nu_{\max} + 1) \log\left(\frac{e\eta_1}{\eta_N}\right),$$

where the final inequality follows from Fact A.3 with $X_n = \alpha_n$ and the identity $\eta_m = 1/(\sum_{k=1}^m \alpha_k)$.

Using a different approach, we can also write

$$H_\eta \leqslant \sum_{m=1}^N \alpha_m \sum_{n=m}^{m+\nu_m} \eta_n \leqslant \left(\sum_{m=1}^N \alpha_m\right) \max_{n\in[N]} \sum_{m=n}^{n+\nu_n} \eta_m = \frac{\max_{n\in[N]} \sum_{m=n}^{n+\nu_n} \eta_m}{\eta_N}.$$

This concludes the proof. $\qquad\square$

## C.3. Proofs of Theorem 4.1 and Lemma 4.2

**Theorem 4.1** (Restated)**.** *Predictions $\{z_n\}_{n=1}^N$ generated by either P-FTRL* (4) *or OMD* (5) *satisfy*

$$\mathfrak{R}_N(u) \leqslant \sum_{n=1}^N \frac{1/\eta_n - 1/\eta_{n-1}}{2} \|z_n - u\|^2 + \frac{1}{2}\sum_{n=1}^N \eta_n \|c_n\|_\star^2 \quad and \quad \mathfrak{D}_N \leqslant \sum_{n=1}^N \eta_n \nu_n \|c_n\|_\star.$$

*Proof.* The regret bound for P-FTRL follows from Theorem C.1 with regularizer functions $\psi_n(z) = \sum_{m=1}^n \frac{1/\eta_m - 1/\eta_{m-1}}{2}\|z_m - z\|^2$. The regret bound for OMD follows from Theorem C.4 with $\psi(z) = \frac{1}{2}\|z\|^2$ so that $B_\psi(x; z) = \frac{1}{2}\|x - z\|^2$.

For both algorithms, it holds that $\|z_{n+1} - z_n\| \leqslant \eta_n \|c_n\|_\star$ for all $n \in [N-1]$, as shown in Theorem C.1 and Lemma C.5. Using triangle inequality and the fact that $\nu_m = |\{n : n - \lambda_n \leqslant m < n\}|$, we can write

$$\mathfrak{D}_N = \sum_{n=1}^N \|z_n - z_{n-\lambda_n}\|$$

$$\leqslant \sum_{n=1}^N \sum_{m=1}^N \mathbb{I}(n - \lambda_n \leqslant m < n)\|z_{m+1} - z_m\|$$

$$= \sum_{m=1}^N (\sum_{n=1}^N \mathbb{I}(n - \lambda_n \leqslant m < n))\|z_{m+1} - z_m\|$$

$$= \sum_{m=1}^N \nu_m \|z_{m+1} - z_m\|$$

$$\leqslant \sum_{m=1}^N \eta_m \nu_m \|c_m\|_\star,$$

where the final inequality plugs-in $\|z_{m+1} - z_m\| \leqslant \eta_m \|c_m\|_\star$ for single-step drifts. $\qquad\square$

**Lemma 4.2** (Restated)**.** *Predictions $\{z_n\}_{n=1}^N$ generated by P-FTRL* (4) *satisfy*

$$\mathfrak{D}_N \leqslant \sum_{n=1}^N \eta_n \|\sum_{m=n-\lambda_n}^{n-1} c_m\|_\star + D\,H_\eta \quad where \quad H_\eta = \sum_{n=1}^N (1 - \eta_n/\eta_{n-\lambda_n}).$$

*Proof.* From Lemma C.5, we write

$$\mathfrak{D}_N = \sum_{n=1}^N \|z_n - z_{n-\lambda_n}\|$$

$$\leqslant \sum_{n=1}^N \left(\eta_n \|\sum_{m=n-\lambda_n}^{n-1} c_m\|_\star + (1 - \eta_n/\eta_{n-\lambda_n})D\right)$$

$$= \sum_{n=1}^N \eta_n \|\sum_{m=n-\lambda_n}^{n-1} c_m\|_\star + DH_\eta. \qquad\square$$

## D. Guarantees for OCO and BCO with Delays: Proofs

### D.1. Proof of Theorem 5.2

**Theorem 5.2** (Restated). *For any base algorithm $\mathcal{B}$ and comparator $u \in \mathcal{K}$, $\mathcal{W}_{\mathrm{OCO}}(\mathcal{B})$ guarantees*

$$R_T(u) \leqslant \mathfrak{R}_T^{\mathrm{drift}}(u; G),$$

*where $\mathfrak{R}_T^{\mathrm{drift}}(u; W)$ is the drift-penalized regret of $\mathcal{B}$ with loss vectors $c_n = \widetilde{g}_n$ and lags $\lambda_n = \widetilde{d}_n^\star$.*

*Under $\lambda$-strong convexity (2.3), it further holds that*

$$R_T(u) \leqslant \mathfrak{R}_T^{\mathrm{drift}}(u; 3G) - \tfrac{\lambda}{2} \sum_{n=1}^T \|z_n - u\|^2.$$

*Proof.* The drift-penalized regret of $\mathcal{B}$ for these loss vectors and lags is defined as

$$\mathfrak{R}_T^{\mathrm{drift}}(u; W) = \sum_{n=1}^T \langle \widetilde{g}_n, z_n - u \rangle + W \sum_{n=1}^T \|z_{n-\widetilde{d}_n^\star} - z_n\|.$$

We treat the general convex case as 0-strongly convex. For $\lambda \geqslant 0$, the $\lambda$-strong convexity of the loss functions $f_t$ gives

$$
\begin{aligned}
\sum_{t=1}^T (f_t(x_t) - f_t(u)) &\leqslant \sum_{t=1}^T \langle g_t, x_t - u \rangle - \tfrac{\lambda}{2} \sum_{t=1}^T \|x_t - u\|^2 \\
&\overset{(a)}{=} \sum_{n=1}^T \langle \widetilde{g}_n, \widetilde{x}_n - u \rangle - \tfrac{\lambda}{2} \sum_{n=1}^T \|\widetilde{x}_n - u\|^2 \\
&\overset{(b)}{=} \sum_{n=1}^T \langle \widetilde{g}_n, z_{n-\widetilde{d}_n^\star} - u \rangle - \tfrac{\lambda}{2} \sum_{n=1}^T \|z_{n-\widetilde{d}_n^\star} - u\|^2 \\
&\leqslant \sum_{n=1}^T \langle \widetilde{g}_n, z_n - u \rangle + \sum_{n=1}^T \|\widetilde{g}_n\|_\star \|z_{n-\widetilde{d}_n^\star} - z_n\| - \tfrac{\lambda}{2} \sum_{n=1}^T \|z_{n-\widetilde{d}_n^\star} - u\|^2 \\
&\overset{(c)}{\leqslant} \sum_{n=1}^T \langle \widetilde{g}_n, z_n - u \rangle + G \sum_{n=1}^T \|z_{n-\widetilde{d}_n^\star} - z_n\| - \tfrac{\lambda}{2} \sum_{n=1}^T \|z_{n-\widetilde{d}_n^\star} - u\|^2,
\end{aligned}
$$

where (a) replaces sequences $\{x_t\}_{t=1}^T$, $\{g_t\}_{t=1}^T$ with their observation-orderings (Definition 3.5), (b) substitutes $\widetilde{x}_n = z_{n-\widetilde{d}_n^\star}$ which follows from Lemma 3.6, and (c) applies the gradient norm bound (Assumption 2.2).

For the general convex case ($\lambda = 0$), this immediately gives $R_T(u) \leqslant \mathfrak{R}_T^{\mathrm{drift}}(u; G)$.

For the strongly convex case ($\lambda > 0$), we convert the distance terms from $z_{n-\widetilde{d}_n^\star}$ to $z_n$. Since $\|z_{n-\widetilde{d}_n^\star} - u\|, \|z_n - u\| \leqslant D \leqslant \tfrac{2G}{\lambda}$ by Fact A.1, we have

$$\tfrac{\lambda}{2}(\|z_n - u\|^2 - \|z_{n-\widetilde{d}_n^\star} - u\|^2) = \tfrac{\lambda}{2} \langle z_{n-\widetilde{d}_n^\star} + z_n - 2u, z_n - z_{n-\widetilde{d}_n^\star} \rangle \leqslant 2G \|z_{n-\widetilde{d}_n^\star} - z_n\|.$$

Summing over $n$ and rearranging yields $R_T(u) \leqslant \mathfrak{R}_T^{\mathrm{drift}}(u; 3G) - \tfrac{\lambda}{2} \sum_{n=1}^T \|z_n - u\|^2$. $\qquad \square$

### D.2. Proofs of Corollaries 5.3 and 5.4

**Theorem D.1.** $\mathcal{W}_{\mathrm{OCO}}(\mathcal{B})$, where $\mathcal{B}$ executes P-FTRL (4) updates, guarantees that in the

$$\text{general convex case}: \quad R_T(u) \leqslant \tfrac{D^2}{2\eta_T} + G^2 \sum_{n=1}^T \eta_n (\widetilde{\sigma}_n^\star + 1),$$

$$\text{strongly convex case (2.3)}: \quad R_T(u) \leqslant \sum_{n=1}^T \tfrac{1/\eta_n - 1/\eta_{n-1} - \lambda}{2} \|z_n - u\|^2 + 3G^2 \sum_{n=1}^T \eta_n(\widetilde{d}_n^\star + 1) + 3GDH_\eta,$$

*where $H_\eta = \sum_{n=1}^T (1 - \eta_n/\eta_{n-\widetilde{d}_n^\star})$ denotes the total learning rate misalignment.*

*Proof.* Both results follow from Theorem 5.2 once we bound the standard regret $\mathfrak{R}_T(u) = \sum_{n=1}^T \langle \widetilde{g}_n, z_n - u \rangle$ and prediction drift $\mathfrak{D}_T = \sum_{n=1}^T \|z_{n-\widetilde{d}_n^\star} - z_n\|$ components.

To control $\mathfrak{R}_T(u)$ in both cases, apply Theorem 4.1 for $\|\widetilde{g}_n\|_\star \leqslant G$ and $\|z_n - u\| \leqslant D$, as follows

$$\mathfrak{R}_T(u) \leqslant \sum_{n=1}^T \tfrac{1/\eta_n - 1/\eta_{n-1}}{2} \|z_m - u\|^2 + \tfrac{1}{2} \sum_{n=1}^T \eta_n \|g_n\|_\star^2 \leqslant \tfrac{D^2}{2\eta_T} + G^2 \sum_{n=1}^T \eta_n.$$

To control $\mathfrak{D}_T$ in the general convex case, use Theorem 4.1 to write

$$\mathfrak{D}_T \leqslant \sum_{n=1}^{T} \eta_n \widetilde{\sigma}_n^{\star} \|\widetilde{g}_n\|_{\star} \leqslant G \sum_{n=1}^{T} \eta_n \widetilde{\sigma}_n^{\star}.$$

Then, the general convex case follows from the corresponding inequality in Theorem 5.2:

$$R_T(u) \leqslant \mathfrak{R}_T(u) + G \mathfrak{D}_T \leqslant \tfrac{D^2}{2\eta_T} + G^2 \sum_{n=1}^{T} \eta_n + G^2 \sum_{n=1}^{T} \eta_n \widetilde{\sigma}_n^{\star} \leqslant \tfrac{D^2}{2\eta_T} + G^2 \sum_{n=1}^{T} \eta_n (\widetilde{\sigma}_n^{\star} + 1).$$

To control $\mathfrak{D}_T$ in the strongly convex case, use Lemma 4.2 to write

$$\mathfrak{D}_T \leqslant \sum_{n=1}^{T} \eta_n \| \sum_{m=n-\widetilde{d}_n^{\star}}^{n-1} \widetilde{g}_m \|_{\star} + D\, H_\eta \leqslant G \sum_{n=1}^{T} \eta_n \widetilde{d}_n^{\star} + D\, H_\eta,$$

where $H_\eta = \sum_{n=1}^{T}(1 - \eta_n/\eta_{n-\widetilde{d}_n^{\star}})$ in this case with lags $\lambda_n = \widetilde{d}_n^{\star}$.

Finally, the strongly convex case follows from the corresponding inequality in Theorem 5.2:

$$\begin{aligned}
R_T(u) &\leqslant \mathfrak{R}_T(u) - \tfrac{\lambda}{2} \sum_{n=1}^{T} \|z_n - u\|^2 + 3G\, \mathfrak{D}_T \\
&\leqslant \sum_{n=1}^{T} \tfrac{1/\eta_n - 1/\eta_{n-1} - \lambda}{2} \|z_n - u\|^2 + G^2 \sum_{n=1}^{T} \eta_n + 3G^2 \sum_{n=1}^{T} \eta_n \widetilde{d}_n^{\star} + 3GDH_\eta \\
&\leqslant \sum_{n=1}^{T} \tfrac{1/\eta_n - 1/\eta_{n-1} - \lambda}{2} \|z_n - u\|^2 + 3G^2 \sum_{n=1}^{T} \eta_n (\widetilde{d}_n^{\star} + 1) + 3GDH_\eta.
\end{aligned}$$

This concludes the proof for both cases. □

**Theorem D.2.** $\mathcal{W}_{\mathrm{OCO}}(\mathcal{B})$, *where $\mathcal{B}$ executes OMD (5) updates, guarantees that in the*

$$\text{general convex convex: } R_T(u) \leqslant \tfrac{D^2}{2\eta_T} + G^2 \sum_{n=1}^{T} \eta_n (\widetilde{\sigma}_n^{\star} + 1),$$

$$\text{strongly convex case (2.3): } R_T(u) \leqslant \sum_{n=1}^{T} \tfrac{1/\eta_n - 1/\eta_{n-1} - \lambda}{2} \|z_n - u\|^2 + 3G^2 \sum_{n=1}^{T} \eta_n (\widetilde{\sigma}_n^{\star} + 1).$$

*Proof.* As in the proof of Theorem D.1, we use Theorem 4.1 to show that

$$\mathfrak{R}_T(u) \leqslant \sum_{n=1}^{T} \tfrac{1/\eta_n - 1/\eta_{n-1}}{2} \|z_m - u\|^2 + G^2 \sum_{n=1}^{T} \eta_n \quad \text{and} \quad \mathfrak{D}_T \leqslant G \sum_{n=1}^{T} \eta_n \widetilde{\sigma}_n^{\star}.$$

The general convex case similarly follows from its corresponding inequality in Theorem 5.2:

$$R_T(u) \leqslant \mathfrak{R}_T(u) + G \mathfrak{D}_T \leqslant \tfrac{D^2}{2\eta_T} + G^2 \sum_{n=1}^{T} \eta_n + G^2 \sum_{n=1}^{T} \eta_n \widetilde{\sigma}_n^{\star} \leqslant \tfrac{D^2}{2\eta_T} + G^2 \sum_{n=1}^{T} \eta_n (\widetilde{\sigma}_n^{\star} + 1).$$

The strongly convex case also follows from its corresponding inequality in Theorem 5.2:

$$\begin{aligned}
R_T(u) &\leqslant \mathfrak{R}_T(u) - \tfrac{\lambda}{2} \sum_{n=1}^{T} \|z_n - u\|^2 + 3G\, \mathfrak{D}_T \\
&\leqslant \sum_{n=1}^{T} \tfrac{1/\eta_n - 1/\eta_{n-1} - \lambda}{2} \|z_n - u\|^2 + G^2 \sum_{n=1}^{T} \eta_n + 3G^2 \sum_{n=1}^{T} \eta_n \widetilde{\sigma}_n^{\star} \\
&\leqslant \sum_{n=1}^{T} \tfrac{1/\eta_n - 1/\eta_{n-1} - \lambda}{2} \|z_n - u\|^2 + 3G^2 \sum_{n=1}^{T} \eta_n (\widetilde{\sigma}_n^{\star} + 1).
\end{aligned}$$

This concludes the proof for both cases. □

**Corollary 5.3** (Restated). $\mathcal{W}_{\mathrm{OCO}}(\mathcal{B})$, *where $\mathcal{B}$ runs P-FTRL (4) or OMD (5) with $\eta_n = \frac{D/G}{\sqrt{n + \sum_{m=1}^{n} \widetilde{\sigma}_m^{\star}}}$, guarantees*

$$R_T(x^*) = O\left( GD \sqrt{T + d_{tot}} \right).$$

*Proof.* Using the general convex case of Theorem D.1 for P-FTRL and Theorem D.2 for OMD, we can write

$$\begin{aligned}
R_T(x^*) &\leqslant \tfrac{D^2}{2\eta_T} + G^2 \sum_{n=1}^{T} \eta_n (\widetilde{\sigma}_n^{\star} + 1) \\
&\overset{(a)}{\leqslant} GD \sqrt{T + \sum_{n=1}^{T} \widetilde{\sigma}_n^{\star}} + 2GD \sqrt{\sum_{n=1}^{T} (\widetilde{\sigma}_n^{\star} + 1)} \\
&\overset{(b)}{=} 3GD \sqrt{T + d_{\mathrm{tot}}},
\end{aligned}$$

where (a) substitutes learning rate values and applies Fact A.4 for $X_n = \widetilde{\sigma}_n^{\star} + 1$ and (b) uses the indentity $\sum_{t=1}^{T} \sigma_t^{\star} = d_{\mathrm{tot}}$ from Theorem 3.4. □

**Corollary 5.4** (Restated). *Under $\lambda$-strong convexity (2.3), $\mathcal{W}_{\mathrm{OCO}}(\mathcal{B})$, where $\mathcal{B}$ runs P-FTRL (4) with $\eta_n = \frac{1}{n\lambda}$, guarantees*

$$R_T(x^*) = O\left(\tfrac{G^2}{\lambda} \min\left\{(\sigma_{\max}+1)\log T, \sqrt{T+d_{tot}}\right\}\right).$$

*Running OMD (5) with $\eta_n = \frac{1}{n\lambda}$ instead guarantees*

$$R_T(x^*) = O\left(\tfrac{G^2}{\lambda}(\sigma_{\max}+1)\log T\right).$$

*Proof.* We prove the result for P-FTRL first. For learning rates $\eta_n = \frac{1}{n\lambda}$, using inequality $D \leqslant \frac{2G}{\lambda}$ from Fact A.1, term $H_\eta = \sum_{n=1}^T \eta_n(1/\eta_n - 1/\eta_{n-\widetilde{d}_n^\star})$ can be bounded from above as

$$H_\eta = \sum_{n=1}^T \eta_n(\lambda n - \lambda(n - \widetilde{d}_n^\star)) = \lambda \sum_{n=1}^T \eta_n \widetilde{d}_n^\star \leqslant (2G/D) \sum_{n=1}^T \eta_n \widetilde{d}_n^\star.$$

Then, using the strongly convex case of Theorem D.1, we write

$$\begin{aligned}
R_T(x^*) &\leqslant \sum_{n=1}^T \tfrac{1/\eta_n - 1/\eta_{n-1} - \lambda}{2} \|z_n - x^*\|^2 + 3G^2 \sum_{n=1}^T \eta_n(\widetilde{d}_n^\star + 1) + 3GDH_\eta \\
&\overset{(a)}{\leqslant} 0 + 3G^2 \sum_{n=1}^T \eta_n(\widetilde{d}_n^\star + 1) + 6G^2 \sum_{n=1}^T \eta_n \widetilde{d}_n^\star \\
&\overset{(b)}{\leqslant} 9G^2 \sum_{n=1}^T \tfrac{\widetilde{d}_n^\star + 1}{\lambda n} \\
&\overset{(c)}{\leqslant} \tfrac{9G^2}{\lambda}\left(\min\{\sigma_{\max}\log(eT), 2\sqrt{d_{\mathrm{tot}}}\} + \log(eT)\right),
\end{aligned}$$

where (a) substitutes learning rate values in the first term and applies the above bound on $H_\eta$, (b) substitutes $\eta_n = 1/(\lambda n)$, and (c) uses Fact B.4 for $\widetilde{d}_n^\star$ and Fact A.3 to get the desired bound.

For OMD, using the strongly convex case of Theorem D.2, we write

$$\begin{aligned}
R_T(x^*) &\leqslant \sum_{n=1}^T \tfrac{1/\eta_n - 1/\eta_{n-1} - \lambda}{2} \|z_n - x^*\|^2 + 3G^2 \sum_{n=1}^T \eta_n(\widetilde{\sigma}_n^\star + 1) \\
&\overset{(a)}{\leqslant} 0 + 3G^2 \sum_{n=1}^T \tfrac{\widetilde{\sigma}_n^\star + 1}{\lambda n} \\
&\overset{(b)}{\leqslant} \tfrac{3G^2}{\lambda}\left(\sigma_{\max}\log(eT) + \log(eT)\right),
\end{aligned}$$

where (a) substitutes learning rate values and (b) uses Fact B.4 for $\widetilde{\sigma}_n^\star$ to get the final bound. $\qquad\square$

### D.3. Proof of Theorem 5.6

In this section, we prove Theorem 5.6. We begin by recalling notation from the main body.

Algorithm $\mathcal{B}$ generates base-predictions $\{z_n\}_{n=1}^T$, and in round $t$ the wrapper $\mathcal{W}_{BCO}(\mathcal{B})$ produces prediction $x_t = (1 - \delta_t/r)\, y_t + \delta_t u_t$, where $y_t = \bar{z}$ is the current base-prediction and $u_t \sim \mathrm{Unif}(\mathbb{S}^{k-1})$ is sampled independently. Define the shrunk suggestions $y_t^\delta = (1 - \delta_t/r)\, y_t \in (1 - \delta_t/r)\mathcal{K}$, so that $x_t = y_t^\delta + \delta_t u_t$.

Under observation-ordering (Definition 3.5), these quantities become

$$\widetilde{y}_n = z_{n-\widetilde{d}_n^\star}, \quad \widetilde{y}_n^\delta = (1 - \tfrac{\widetilde{\delta}_n}{r})\, z_{n-\widetilde{d}_n^\star}, \quad \widetilde{x}_n = \widetilde{y}_n^\delta + \widetilde{\delta}_n \widetilde{u}_n.$$

For the smoothed losses $\widetilde{f}_n^\delta(x) = \mathbb{E}_{v \sim \mathrm{Unif}(\mathbb{B}^k)}[\widetilde{f}_n(x + \widetilde{\delta}_n v)]$ defined on $(1 - \tfrac{\widetilde{\delta}_n}{r})\mathcal{K}$, let $\widetilde{g}_n^\delta = \nabla \widetilde{f}_n^\delta(\widetilde{y}_n^\delta)$.
By Theorem 5.5, the single-point estimator $\widetilde{\widehat{g}}_n^\delta = \frac{k}{\widetilde{\delta}_n} \widetilde{f}_n(\widetilde{x}_n)\widetilde{u}_n$ satisfies $\mathbb{E}[\widetilde{\widehat{g}}_n^\delta \mid z_{n-\widetilde{d}_n^\star}] = \widetilde{g}_n^\delta$.

**Theorem 5.6** (Restated). *For any base-algorithm $\mathcal{B}$, non-increasing sequence $\{\delta_t\}_{t\geqslant 1} \subset (0, r]$, and comparator $u \in \mathcal{K}$, $\mathcal{W}_{\mathrm{BCO}}(\mathcal{B})$ guarantees*

$$\bar{R}_T(u) \leqslant \mathbb{E}\left[\mathfrak{R}_T^{\mathrm{drift}}(u; G)\right] + \tfrac{6GR\delta_{\mathrm{tot}}}{r},$$

where $\mathfrak{R}_T^{\text{drift}}(u; W)$ is the drift-penalized regret of $\mathcal{B}$ on loss vectors $c_n = \widehat{\widetilde{g}}_n^\delta$ with lags $\lambda_n = \widetilde{d}_n^\star$, and $\delta_{\text{tot}} = \sum_{t=1}^T \delta_t$. Under $\lambda$-strong convexity (2.3), it further holds that

$$\bar{R}_T(u) \leqslant \mathbb{E}\Big[\mathfrak{R}_T^{\text{drift}}(u; 3G) - \tfrac{\lambda}{2}\sum_{n=1}^T \|z_n - u\|^2\Big] + \tfrac{10GR\delta_{\text{tot}}}{r}.$$

*Proof.* Fix comparator $v \in \mathcal{K}$. For $n \in [T]$, let $\widetilde{v}_n^\delta = (1 - \widetilde{\delta}_n/r)v$ which satisfies $\|\widetilde{v}_n^\delta - v\| = \frac{\widetilde{\delta}_n\|v\|}{r} \leqslant \frac{R\widetilde{\delta}_n}{r}$. This gives

$$\bar{R}_T(v) = \mathbb{E}\left[\sum_{n=1}^T (\widetilde{f}_n(\widetilde{x}_n) - \widetilde{f}_n(v))\right]$$

$$= \mathbb{E}\left[\sum_{n=1}^T (\widetilde{f}_n^\delta(\widetilde{y}_n^\delta) - \widetilde{f}_n^\delta(\widetilde{v}_n^\delta))\right]$$

$$+ \mathbb{E}\left[\sum_{n=1}^T (\widetilde{f}_n(\widetilde{x}_n) - \widetilde{f}_n(\widetilde{y}_n^\delta))\right] + \mathbb{E}\left[\sum_{n=1}^T (\widetilde{f}_n(\widetilde{y}_n^\delta) - \widetilde{f}_n^\delta(\widetilde{y}_n^\delta))\right]$$

$$+ \mathbb{E}\left[\sum_{n=1}^T (\widetilde{f}_n^\delta(\widetilde{v}_n^\delta) - \widetilde{f}_n(\widetilde{v}_n^\delta))\right] + \mathbb{E}\left[\sum_{n=1}^T (\widetilde{f}_n(\widetilde{v}_n^\delta) - \widetilde{f}_n(v))\right]$$

$$\overset{(a)}{\leqslant} \mathbb{E}\left[\sum_{n=1}^T (\widetilde{f}_n^\delta(\widetilde{y}_n^\delta) - \widetilde{f}_n^\delta(\widetilde{v}_n^\delta))\right] + \sum_{n=1}^T \left(G\widetilde{\delta}_n + G\widetilde{\delta}_n + G\widetilde{\delta}_n + \tfrac{GR}{r}\widetilde{\delta}_n\right)$$

$$\overset{(b)}{\leqslant} \mathbb{E}\left[\sum_{n=1}^T \left(\langle \widetilde{g}_n^\delta,\, \widetilde{y}_n^\delta - \widetilde{v}_n^\delta\rangle - \tfrac{\lambda}{2}\|\widetilde{y}_n^\delta - \widetilde{v}_n^\delta\|^2\right)\right] + \tfrac{4GR\delta_{\text{tot}}}{r}$$

$$\overset{(c)}{=} \mathbb{E}\left[\sum_{n=1}^T \left((1 - \tfrac{\widetilde{\delta}_n}{r})\langle \widetilde{g}_n^\delta,\, z_{n - \widetilde{d}_n^\star} - v\rangle - \tfrac{\lambda}{2}(1 - \tfrac{\widetilde{\delta}_n}{r})^2 \|z_{n - \widetilde{d}_n^\star} - v\|^2\right)\right] + \tfrac{4GR\delta_{\text{tot}}}{r}$$

$$= \mathbb{E}\left[\sum_{n=1}^T \left(\langle \widetilde{g}_n^\delta,\, z_n - v\rangle - \tfrac{\lambda}{2}\|z_{n - \widetilde{d}_n^\star} - v\|^2 + \langle \widetilde{g}_n^\delta,\, z_{n - \widetilde{d}_n^\star} - z_n\rangle\right)\right] + \tfrac{4GR\delta_{\text{tot}}}{r}$$

$$+ \mathbb{E}\left[\sum_{n=1}^T (\tfrac{\widetilde{\delta}_n}{r}\langle \widetilde{g}_n^\delta,\, v - z_{n - \widetilde{d}_n^\star}\rangle + (\tfrac{\lambda\widetilde{\delta}_n}{r} - \tfrac{\lambda\widetilde{\delta}_n^2}{2r^2})\|z_{n - \widetilde{d}_n^\star} - v\|^2)\right]. \tag{6}$$

where (a) uses $G$-Lipschitzness of $\widetilde{f}_n$ and Theorem 5.5 to bound the last four terms in the previous line, (b) uses $\lambda$-strong convexity of $\widetilde{f}_n^\delta$, and (c) substitutes $\widetilde{y}_n^\delta = (1 - \tfrac{\widetilde{\delta}_n}{r})z_{n - \widetilde{d}_n^\star}$ and $\widetilde{v}_n^\delta = (1 - \tfrac{\widetilde{\delta}_n}{r})v$.

The final term in (6) can be bound using Cauchy-Schwartz inequality as follows:

$$\mathbb{E}\left[\sum_{n=1}^T (\tfrac{\widetilde{\delta}_n}{r}\langle \widetilde{g}_n^\delta,\, v - z_{n - \widetilde{d}_n^\star}\rangle + (\tfrac{\lambda\widetilde{\delta}_n}{r} - \tfrac{\lambda\widetilde{\delta}_n^2}{2r^2})\|z_{n - \widetilde{d}_n^\star} - v\|^2)\right]$$

$$\leqslant \mathbb{E}\left[\sum_{n=1}^T (\tfrac{\widetilde{\delta}_n}{r}\|\widetilde{g}_n^\delta\|_\star\|z_{n - \widetilde{d}_n^\star} - v\| + \tfrac{\lambda\widetilde{\delta}_n}{r}\|z_{n - \widetilde{d}_n^\star} - v\|^2)\right]$$

$$\overset{(d)}{\leqslant} \sum_{n=1}^T \left(\tfrac{GD\widetilde{\delta}_n}{r} + \tfrac{\lambda D^2\widetilde{\delta}_n}{r}\right) = \tfrac{(G + \lambda D)D\delta_{\text{tot}}}{r}, \tag{7}$$

where (d) follows from the facts that $\|z_{n - \widetilde{d}_n^\star} - v\| \leqslant D$ and $\|\widetilde{g}_n^\delta\|_\star \leqslant G$ due to Theorem 5.5.

For every $n \in [T]$, the direction $\widetilde{u}_n$ and base-prediction $z_n$ are independent: $\widetilde{u}_n$ is sampled when making the $n$-th prediction and only affects the $n$-th observation $\widetilde{f}_n(\widetilde{x}_n) = \widetilde{f}_n((1 - \widetilde{\delta}_n/r)\, z_{n - \widetilde{d}_n^\star} + \widetilde{\delta}_n\widetilde{u}_n)$, whereas $z_n$ is the output of $\mathcal{B}$ after processing the $(n - 1)$-th observation. By Theorem 5.5,

$$\mathbb{E}[\widehat{\widetilde{g}}_n^\delta \mid z_n] = \mathbb{E}\left[\tfrac{k}{\widetilde{\delta}_n}\widetilde{f}_n(\widetilde{x}_n)\widetilde{u}_n \mid z_n\right] = \nabla \widetilde{f}_n^\delta\left((1 - \tfrac{\widetilde{\delta}_n}{r})\, z_{n - \widetilde{d}_n^\star}\right) = \widetilde{g}_n^\delta.$$

Hence, the expected regret term $\mathbb{E}[\mathfrak{R}(u)]$ satisfies

$$\mathbb{E}[\mathfrak{R}(u)] = \mathbb{E}\left[\sum_{n=1}^T \langle \widehat{\widetilde{g}}_n^\delta,\, z_n - u\rangle\right] \overset{(e)}{=} \mathbb{E}\left[\sum_{n=1}^T \langle \widetilde{g}_n^\delta,\, z_n - u\rangle\right]. \tag{8}$$

Combining (6), (7), (8) and the facts that $D \leqslant 2R$ and $\langle \widetilde{g}_n^\delta,\, z_{n - \widetilde{d}_n^\star} - z_n\rangle \leqslant G\|z_{n - \widetilde{d}_n^\star} - z_n\|$, we have

$$\bar{R}_T(v) \leqslant \mathbb{E}\left[\mathfrak{R}_T(v) - \tfrac{\lambda}{2}\sum_{n=1}^T \|z_{n - \widetilde{d}_n^\star} - v\|^2 + G\mathfrak{D}_T\right] + \tfrac{(6G + 2\lambda D)R\delta_{\text{tot}}}{r}. \tag{9}$$

This concludes the proof for the convex case ($\lambda = 0$). It remains to consider the strongly convex case ($\lambda > 0$).

For all $n \in [T]$,

$$\tfrac{\lambda}{2}\left(\|z_n - u\|^2 - \|z_{n-\tilde{d}_n^\star} - u\|^2\right) = \tfrac{\lambda}{2}\langle z_{n-\tilde{d}_n^\star} + z_n - 2u,\, z_n - z_{n-\tilde{d}_n^\star}\rangle \leqslant \lambda D\,\|z_{n-\tilde{d}_n^\star} - z_n\|.$$

Combining with (9) and substituting the bound $D \leqslant \frac{2G}{\lambda}$, from Fact A.1, yields

$$\bar{R}_T(u) \leqslant \mathbb{E}\left[\mathfrak{R}_T(u) - \tfrac{\lambda}{2}\sum_{n=1}^T \|z_n - u\|^2 + 3G\,\mathfrak{D}_T\right] + \tfrac{10GR\delta_{\text{tot}}}{r}.$$

This completes the proof. $\square$

## D.4. Proofs of Corollaries 5.7 and 5.8

**Theorem D.3.** $\mathcal{W}_{\text{BCO}}(\mathcal{B})$, *where $\mathcal{B}$ executes P-FTRL* (4) *updates, guarantees that in the*

*general convex convex*: $\bar{R}_T(u) \leqslant \frac{D^2}{2\eta_T} + \sum_{n=1}^T \eta_n\left(G\,\mathbb{E}[\|\Gamma_n\|] + (kM/\tilde{\delta}_n)^2\right) + GDH_\eta + \frac{6GR\delta_{\text{tot}}}{r}$,

*strongly convex case (2.3)*: $\bar{R}_T(u) \leqslant \sum_{n=1}^T \frac{1/\eta_n - 1/\eta_{n-1} - \lambda}{2}\mathbb{E}[\|z_n - u\|^2]$
$$+ \sum_{n=1}^T \eta_n\left(3G\,\mathbb{E}[\|\Gamma_n\|] + (kM/\tilde{\delta}_n)^2\right) + 3GDH_\eta + \frac{10GR\delta_{\text{tot}}}{r},$$

*where $\Gamma_n = \sum_{m=n-\tilde{d}_n^\star}^{n-1} \widetilde{\widehat{g}}_m^\delta$ and $H_\eta = \sum_{n=1}^T (1 - \eta_n/\eta_{n-\tilde{d}_n^\star})$.*

*Proof.* Both results follow from Theorem 5.6 once we bound $\mathfrak{R}_T(u)$ and $\mathfrak{D}_T$ components of $\mathfrak{R}_T^{\text{drift}}(u; W)$.

To control $\mathfrak{R}_T(u)$ in both cases, apply Theorem 4.1 for $\|\widetilde{\widehat{g}}_n^\delta\| \leqslant kM/\tilde{\delta}_n$ and $\|z_n - u\| \leqslant D$, as follows

$$\mathfrak{R}_T(u) \leqslant \sum_{n=1}^T \frac{1/\eta_n - 1/\eta_{n-1}}{2}\|z_m - u\|^2 + \tfrac{1}{2}\sum_{n=1}^T \eta_n\|\widetilde{\widehat{g}}_n^\delta\|^2 \leqslant \frac{D^2}{2\eta_T} + \sum_{n=1}^T \eta_n(kM/\tilde{\delta}_n)^2.$$

To control $\mathfrak{D}_T = \sum_{n=1}^T \|z_n - z_{n-\tilde{d}_n^\star}\|$, we use Lemma 4.2 to write

$$\mathfrak{D}_T \leqslant \sum_{n=1}^T \eta_n\left(\|\sum_{m=n-\tilde{d}_n^\star}^{n-1} \tilde{g}_m\|_\star + (1/\eta_n - 1/\eta_{m+1})D\right) \leqslant \sum_{n=1}^T \eta_n\|\Gamma_n\| + DH_\eta.$$

Then, the general convex case follows from the corresponding inequality in Theorem 5.6, as follows:

$$\begin{aligned}
\bar{R}_T(u) &\leqslant \mathbb{E}\left[\mathfrak{R}_T(u) + G\,\mathfrak{D}_T\right] + \tfrac{6GR\delta_{\text{tot}}}{r}\\
&\leqslant \frac{D^2}{2\eta_T} + \sum_{n=1}^T \eta_n(kM/\tilde{\delta}_n)^2 + G\sum_{n=1}^T \eta_n\mathbb{E}[\|\Gamma_n\|] + GDH_\eta + \tfrac{6GR\delta_{\text{tot}}}{r}\\
&= \frac{D^2}{2\eta_T} + \sum_{n=1}^T \eta_n\left(G\,\mathbb{E}[\|\Gamma_n\|] + (kM/\tilde{\delta}_n)^2\right) + GDH_\eta + \tfrac{6GR\delta_{\text{tot}}}{r}.
\end{aligned}$$

The strongly convex case follows from the corresponding inequality in Theorem 5.6:

$$\begin{aligned}
\bar{R}_T(u) &\leqslant \mathbb{E}\left[\mathfrak{R}_T(u) - \tfrac{\lambda}{2}\sum_{n=1}^T \|z_n - u\|^2 + 3G\mathfrak{D}_T\right] + \tfrac{10GR\delta_{\text{tot}}}{r}\\
&\leqslant \sum_{n=1}^T \frac{1/\eta_n - 1/\eta_{n-1} - \lambda}{2}\mathbb{E}[\|z_n - u\|^2] + \sum_{n=1}^T \eta_n(\tfrac{kM}{\tilde{\delta}_n})^2 + 3G\left(\sum_{n=1}^T \eta_n\mathbb{E}[\|\Gamma_n\|] + DH_\eta\right) + \tfrac{10GR\delta_{\text{tot}}}{r}\\
&= \sum_{n=1}^T \frac{1/\eta_n - 1/\eta_{n-1} - \lambda}{2}\mathbb{E}[\|z_n - u\|^2] + \sum_{n=1}^T \eta_n\left(3G\,\mathbb{E}[\|\Gamma_n\|] + (kM/\tilde{\delta}_n)^2\right) + 3GDH_\eta + \tfrac{10GR\delta_{\text{tot}}}{r}.
\end{aligned}$$

This concludes the proof for both cases. $\square$

**Lemma D.4.** *For $\Gamma_n = \sum_{m=n-\tilde{d}_n^\star}^{n-1} \widetilde{\widehat{g}}_m^\delta$ from Theorem D.3 and $\delta_n' = \delta_{\rho(n)+d_{\rho(n)}}$, it holds that*

$$G\,\mathbb{E}\left[\|\Gamma_n\|\right] \leqslant 4(G^2\,\tilde{d}_n^\star + (kM/\delta_n')^2).$$

*Proof.* Note that $\|\tilde{g}_n^\delta\| \le G$ and $\|\widehat{\tilde{g}}_n^\delta\| \le \frac{kM}{\tilde{\delta}_n}$. With probability one, it holds that

$$\|\Gamma_n\| \le \sum_{m=n-\tilde{d}_n^\star}^{n-1} \|\tilde{g}_m^\delta\| + \|\underbrace{\sum_{m=n-\tilde{d}_n^\star}^{n-1} (\widehat{\tilde{g}}_m^\delta - \tilde{g}_m^\delta)}_{=:M_n}\| \le G\tilde{d}_n^\star + \|M_n\|.$$

Since $\delta_n' = \delta_{\rho(n)+d_{\rho(n)}}$, it holds that $\delta_n' \le \delta_t$ for all $t \in \{s : r_s \in I_{\rho(n)}\}$, because for all such $t$ we have $l_t < r_t < r_{\rho(n)} < l_{\rho(n)+d_{\rho(n)}+1}$ (i.e., $t \le \rho(n)+d_{\rho(n)}$) and sequence $(\delta_t)_{t=1}^T$ is non-increasing.

Let $G' = G + kM/\delta_n'$ so that $\|\widehat{g}_t^\delta - g_t^\delta\| \le G'$ for all $t \in \{s : r_s \in I_{\rho(n)}\}$.

Let $v_t = (\widehat{g}_t^\delta - g_t^\delta)\,\mathbb{I}(r_t \in I_{\rho(n)})$ for $t \in [T]$, so that $M_n = \sum_{t=1}^T v_t$, $\mathbb{E}[v_t] = 0$, and $\|v_t\| \le G'\,\mathbb{I}(r_t \in I_{\rho(n)})$. Consider sigma-algebras $\mathcal{F}_t^u = \sigma(u_1, ..., u_{t-1})$. Then, for all $t \in [T]$, $v_t \in \mathcal{F}_{t+1}^u$ and $\mathbb{E}[v_t|\mathcal{F}_t^u] = 0$. Therefore, it holds that

$$\mathbb{E}\left[\|M_n\|^2\right] = \sum_{t=1}^T \mathbb{E}\left[\|v_t\|^2\right] + 2\sum_{t=1}^T \sum_{s=t+1}^T \mathbb{E}[\langle v_t, v_s \rangle]$$

$$\le \sum_{t=1}^T (G')^2\,\mathbb{I}(r_t \in I_{\rho(n)}) + 2\sum_{t=1}^T \sum_{s=t+1}^T \mathbb{E}\left[\langle v_t, \mathbb{E}[v_s|\mathcal{F}_s^u] \rangle\right]$$

$$= (G')^2\,\tilde{d}_n^\star + 0.$$

Combining these results and applying Jensen's inequality, we prove the inequality

$$G\,\mathbb{E}[\|\Gamma_n\|] \le G^2\,\tilde{d}_n^\star + G\sqrt{\mathbb{E}[\|M_n\|^2]} \le G^2\,\tilde{d}_n^\star + G(G + kM/\delta_n')\sqrt{\tilde{d}_n^\star}$$

$$\le 2G^2\,\tilde{d}_n^\star + (G^2\tilde{d}_n^\star)^{1/2}(kM/\delta_n') \le 4(G^2\,\tilde{d}_n^\star + (kM/\delta_n')^2).$$

This concludes the proof of the lemma. $\qquad\square$

**Corollary 5.7** (Restated). $\mathcal{W}_{\text{BCO}}(\mathcal{B})$, *where* $(\delta_t)_{t\ge 1}$ *is set to either a round-dependent schedule* $\delta_t = r\,\min\left\{1, \frac{\sqrt{\nu k}}{t^{1/4}}\right\}$ *or a fixed schedule* $\delta_t = r\,\min\left\{1, \frac{\sqrt{\nu k}}{T^{1/4}}\right\}$ *and* $\mathcal{B}$ *runs P-FTRL* (4) *with* $\eta_n = \frac{D/G}{\sqrt{n + \sum_{m=1}^n (\tilde{\sigma}_m^\star + (\nu kr/\delta_m')^2)}}$, *guarantees*

$$\bar{R}_T(x^*) = O\left(GD\left[\sqrt{d_{tot}} + T^{3/4}\sqrt{\nu k}\right]\right).$$

*Proof.* The proof for a fixed schedule is the same as for a varying schedule, so we only present the latter. From Theorem D.3 and Lemma D.4, we have the following

$$\bar{R}_T(x^*) \le \frac{D^2}{2\eta_T} + \sum_{n=1}^T \eta_n\left(G\,\mathbb{E}[\|\Gamma_n\|] + (kM/\tilde{\delta}_n)^2\right) + GDH_\eta + \frac{6GR\delta_{tot}}{r}$$

$$\le \frac{D^2}{\eta_T} + 5G^2 \sum_{n=1}^T \eta_n\left(\tilde{d}_n^\star + (\nu kr/\delta_n')^2\right) + GDH_\eta + \frac{6GD\delta_{tot}}{r}.$$

We bound the terms $1/\eta_T$, $\sum_{n=1}^T \eta_n\left(\tilde{d}_n^\star + (\nu kr/\delta_n')^2\right)$, and $H_\eta$ separately.

1. As $\delta_T = \min_{t\in[T]} \delta_t$ and $\sum_{n=1}^T \tilde{\sigma}_n^\star = d_{\text{tot}}$, it holds that

$$\frac{1}{\eta_T} = \frac{G}{D}\sqrt{T + \sum_{m=1}^T (\tilde{\sigma}_m^\star + (\nu kr/\delta_m')^2)} \le \frac{G}{D}\left(\sqrt{T + d_{\text{tot}}} + \frac{\nu kr}{\delta_T}\sqrt{T}\right).$$

2. As $\sum_{m=1}^n \tilde{d}_m^\star \le \sum_{m=1}^n \tilde{\sigma}_n^\star$ for all $n \in [T]$ according to Fact B.2, we write

$$\sum_{n=1}^T \eta_n\left(\tilde{d}_n^\star + (\nu kr/\delta_n')^2\right) \le \frac{D}{G}\sum_{n=1}^T \frac{\tilde{d}_n^\star + (\nu kr/\delta_n')^2}{\sqrt{\sum_{m=1}^n (\tilde{d}_m^\star + (\nu kr/\delta_m')^2)}}$$

$$\le \frac{2D}{G}\sqrt{\sum_{n=1}^T (\tilde{d}_n^\star + (\nu kr/\delta_n')^2)}$$

$$\le \frac{2D}{G}\left(\sqrt{T + d_{\text{tot}}} + (\nu kr/\delta_T)\sqrt{T}\right),$$

where the second inequality follows from Fact A.4 with $X_n = \tilde{d}_n^\star + (\nu kr/\delta_n')^2$.

3. For all $n \in [T]$, it holds that

$$\sum_{m=n}^{n+\tilde{\sigma}_n^\star} \eta_m \leqslant \sum_{m=n}^{n+\tilde{\sigma}_n^\star} \frac{D/G}{\sqrt{m+\sum_{k=n}^m \tilde{\sigma}_k}} \overset{(a)}{\leqslant} \sum_{m=0}^{\tilde{\sigma}_n^\star} \frac{D/G}{\sqrt{(m+1)+\sum_{k=0}^m (\tilde{\sigma}_n^\star - k)}} \overset{(b)}{\leqslant} \sum_{m=0}^{\tilde{\sigma}_n^\star} \frac{2D/G}{\sqrt{(m+1)(\tilde{\sigma}_n^\star + 1)}} \overset{(c)}{\leqslant} \frac{4D}{G},$$

where (a) follows from Fact B.3, (b) applies inequality $\sum_{k=0}^m (\tilde{\sigma}_n^\star - k) = \frac{(m+1)(2\tilde{\sigma}_n^\star - m)}{2} \geqslant \frac{(m+1)\tilde{\sigma}_n^\star}{2}$, and (c) uses Fact A.4 to conclude $\sum_{m=0}^{\tilde{\sigma}_n^\star} \frac{1}{\sqrt{m+1}} \leqslant 2\sqrt{\tilde{\sigma}_n^\star + 1}$.

Therefore, by Lemma C.6 and the bound on $1/\eta_T$ above, we have

$$H_\eta \leqslant \frac{\max_{n\in[T]} \sum_{m=n}^{n+\tilde{\sigma}_n^\star} \eta_m}{\eta_T} \leqslant 8 \left( \sqrt{T + d_{\text{tot}}} + (\nu k r/\delta_T)\sqrt{T} \right).$$

Combining all these results, we have

$$\bar{R}_T(x^*) = O\left( GD \left( \sqrt{T + d_{\text{tot}}} + (\nu k r/\delta_T)\sqrt{T} + \delta_{\text{tot}}/r \right) \right).$$

Next, we provide the bounds on $r/\delta_T$ and $\delta_{\text{tot}}/r$ in order to derive the final bound. Trivially, it holds that $r/\delta_T \leqslant \max\left\{1, \frac{T^{1/4}}{\sqrt{\nu k}}\right\}$. Also, note that $\sum_{t=1}^T \frac{1}{t^{1/4}} \leqslant 1 + \int_1^T x^{-1/4}\, dx = 1 + \frac{4}{3}(T^{3/4} - 1) \leqslant \frac{4}{3}T^{3/4}$, so that

$$\delta_{\text{tot}}/r = \sum_{t=1}^T \min\{1, \tfrac{\sqrt{\nu k}}{t^{1/4}}\} \leqslant \sum_{t=1}^T \frac{\sqrt{\nu k}}{t^{1/4}} \leqslant \frac{4}{3}T^{3/4}\sqrt{\nu k}.$$

Thus, we have $r/\delta_T \leqslant \max\left\{1, \frac{T^{1/4}}{\sqrt{\nu k}}\right\}$ and $\delta_{\text{tot}}/r \leqslant \frac{4}{3}T^{3/4}\sqrt{\nu k}$.

Finally, substituting the bounds for $r/\delta_T$ and $\delta_{\text{tot}}/r$, we have

$$\bar{R}_T(x^*) = O\left( GD \left( \sqrt{T + d_{\text{tot}}} + T^{3/4}\sqrt{\nu k} + T^{1/2}(\nu k) \right) \right) = O\left( GD \left( \sqrt{d_{\text{tot}}} + T^{3/4}\sqrt{\nu k} \right) \right).$$

$\square$

**Corollary 5.8** (Restated). *Under $\lambda$-strong convexity (2.3), $\mathcal{W}_{\text{BCO}}(\mathcal{B})$, where $(\delta_t)_{t\geqslant 1}$ is set to either a round-dependent schedule $\delta_t = r \min\left\{1, (\frac{\nu^2 k^2 \log t}{t})^{1/3}\right\}$ or a fixed schedule $\delta_t = r \min\left\{1, (\frac{\nu^2 k^2 \log T}{T})^{1/3}\right\}$ and $\mathcal{B}$ runs P-FTRL (4) with $\eta_n = \frac{1}{n\lambda}$, guarantees*

$$\bar{R}_T(x^*) = O\left( \frac{G^2}{\lambda} \left[ \min\{\sigma_{\max} \log T, \sqrt{d_{tot}}\} + T^{2/3}(\log T)^{1/3}(\nu k)^{2/3} \right] \right).$$

*Proof.* The proof for a fixed schedule is the same as for a varying schedule, so we only present the latter. From Theorem D.3 and Lemma D.4, noting that $1/\eta_n - 1/\eta_{n-1} - \lambda = 0$, we have the following

$$\bar{R}_T(x^*) \leqslant 0 + \sum_{n=1}^T \eta_n \left( 3G\, \mathbb{E}[\|\Gamma_n\|] + (kM/\tilde{\delta}_n)^2 \right) + 3GDH_\eta + \frac{10GR\delta_{\text{tot}}}{r},$$

$$\leqslant 13G^2 \sum_{n=1}^T \eta_n \left( \tilde{d}_n^\star + (\nu k r/\delta_n')^2 \right) + 3GDH_\eta + \frac{10GD\delta_{\text{tot}}}{r}.$$

We bound the terms $\sum_{n=1}^T \eta_n \left( \tilde{d}_n^\star + (\nu k r/\delta_n')^2 \right)$ and $H_\eta$ separately.

1. Using Fact B.4 for dual delays $\tilde{d}_n^\star$ and Fact A.3, we write

$$\sum_{n=1}^T \eta_n \left( \tilde{d}_n^\star + (\nu k r/\delta_n')^2 \right) \leqslant \frac{1}{\lambda} \min\left\{ \sigma_{\max} \log(eT), 2\sqrt{d_{\text{tot}}} \right\} + \frac{1}{\lambda}(\nu k r/\delta_T)^2 \log(eT).$$

2. Substituting learning rate values into $H_\eta$ and applying Fact B.4 for dual delays $\tilde{d}_n^\star$ yields

$$H_\eta = \sum_{n=1}^T (1 - \frac{\eta_n}{\eta_{n-\tilde{d}_n^\star}}) = \sum_{n=1}^T \frac{\tilde{d}_n^\star}{n} \leqslant \min\left\{ \sigma_{\max} \log(eT), 2\sqrt{d_{\text{tot}}} \right\}.$$

Combining all these results and applying the fact $D \leqslant \frac{2G}{\lambda}$ (Fact A.1), we have

$$\bar{R}_T(x^*) \leqslant \left(\tfrac{13G^2}{\lambda} + 3GD\right) \min\left\{\sigma_{\max}\log(eT),\, 2\sqrt{d_{\text{tot}}}\right\} + \tfrac{13G^2}{\lambda}(\nu kr/\delta_T)^2 \log(eT) + \tfrac{10GD\delta_{\text{tot}}}{r}$$

$$= O\left(\tfrac{G^2}{\lambda}\left(\min\left\{\sigma_{\max}\log(T),\, \sqrt{d_{\text{tot}}}\right\} + (\nu kr/\delta_T)^2 \log(T) + \delta_{\text{tot}}/r\right)\right).$$

Next, we provide the bounds on $r/\delta_T$ and $\delta_{\text{tot}}/r$ in order to derive the final bound. Trivially, it holds that $r/\delta_T \leqslant \max\left\{1, (\frac{T}{\nu^2 k^2 \log T})^{1/3}\right\}$. Also, note that $\sum_{t=1}^{T} \frac{1}{t^{1/3}} \leqslant 1 + \int_1^T x^{-1/3}\,dx = 1 + \frac{3}{2}(T^{2/3} - 1) \leqslant \frac{3}{2}T^{2/3}$, so that

$$\delta_{\text{tot}}/r = \sum_{t=1}^{T} \min\left\{1, (\tfrac{\nu^2 k^2 \log t}{t})^{1/3}\right\} \leqslant \sum_{t=1}^{T} \frac{\sqrt{(\nu^2 k^2 \log T)^{1/3}}}{t^{1/3}} \leqslant \tfrac{3}{2}T^{2/3}\log(T)^{1/3}(\nu k)^{2/3}.$$

Thus, we have $r/\delta_T \leqslant \max\left\{1, (\frac{T}{\nu^2 k^2 \log T})^{1/3}\right\}$ and $\delta_{\text{tot}}/r \leqslant \frac{3}{2}T^{2/3}\log(T)^{1/3}(\nu k)^{2/3}$.

Finally, substituting the bounds for $r/\delta_T$ and $\delta_{\text{tot}}/r$, we have

$$\bar{R}_T(x^*) = O\left(\tfrac{G^2}{\lambda}\left(\min\{\sigma_{\max}\log(T),\, \sqrt{d_{\text{tot}}}\} + T^{2/3}\log(T)^{1/3}(\nu k)^{2/3} + \nu^2 k^2 \log(T)\right)\right)$$

$$= O\left(\tfrac{G^2}{\lambda}\left(\min\{\sigma_{\max}\log(T),\, \sqrt{d_{\text{tot}}}\} + T^{2/3}\log(T)^{1/3}(\nu k)^{2/3}\right)\right).$$

$\square$

# E. Skipping Scheme Proofs

**Theorem 6.1** (Restated). *The regret $R_T(u)$ of $\mathcal{S}_{\text{ZS}}(\mathcal{A})$ against any comparator $u \in \mathcal{K}$ satisfies*

$$R_T(u) \leqslant R_T'(u) + GD|R^*|,$$

*where $R_T'(u) = \sum_{t=1}^{T}(f_t'(x_t) - f_t'(u))$ is the regret of $\mathcal{A}$ on delayed OCO problem with losses and delays $\{f_t', d_t'\}_{t=1}^{T}$. Moreover, it holds that*

$$|R^*| + \sqrt{\sum_{t=1}^{T} d_t'} = O\left(\min_{Q \subseteq [T]} \left\{|Q| + \sqrt{\sum_{t \notin Q} d_t}\right\}\right).$$

*Proof.* The regret bound follows directly from the observation that each skipped round contributes at most $GD$ to the regret. Specifically, for all $t \in [T]$, $f_t(x_t) - f_t(u) \leqslant f_t'(x_t) - f_t'(u) + GD \cdot \mathbb{I}(t \in R^*)$. Summing over $t$ yields the first claim.

It remains to establish the second claim regarding the skipped set $R$. We introduce additional notation: let $S_t$ denote the state of the tracking set $S$ at the start of round $t$. By construction, $d_t' = \sum_{s=1}^{T} \mathbb{I}(t \in S_s) \leqslant d_t$. Then, $d_{\text{tot}}' = \sum_{t=1}^{T} d_t' = \sum_{t=1}^{T} |S_t|$. As $\mathcal{D}_t = \mathcal{D}_{t-1} + |S_t|$, it also holds that $d_{\text{tot}}' = \mathcal{D}_T$. Furthermore, by the preemption rule, round $t$ is not removed before round $t + d_t' - 1$, which implies $d_t' \leqslant \sqrt{\mathcal{D}_{t-1}} + 1 \leqslant \sqrt{d_{\text{tot}}'} + 1$ for all $t \in [T]$.

**Part 1 (Bounding $|R^*|$):** We show that $|R^*| \leqslant 2\sqrt{d_{\text{tot}}'}$. Let $R^* = \{t_1, \ldots, t_{|R^*|}\}$ ordered by time of skipping. We claim that $d_{t_s}' \geqslant s/2$ for all $s \in [|R^*|]$. Indeed, for each $s \in [|R^*|]$,

$$d_{t_s}' > \sqrt{\mathcal{D}_{t_s + d_{t_s}'}} = \sqrt{\sum_{t=1}^{T} \min\{d_t', t_s + d_{t_s}' - t\}} \geqslant \sqrt{\sum_{k=1}^{s} d_{t_k}'},$$

which rearranges to $d_{t_s}' \geqslant \frac{1 + \sqrt{1 + 4\sum_{k=1}^{s-1} d_{t_k}'}}{2}$ by solving the quadratic inequaltiy.

We proceed by induction. The base case gives $d_{t_1}' \geqslant 1$. Assuming $d_{t_k}' \geqslant k/2$ for all $k < s$, we obtain

$$d_{t_s}' \geqslant \frac{1 + \sqrt{1 + 4\sum_{k=1}^{s-1} \frac{k}{2}}}{2} = \frac{1 + \sqrt{1 + s(s-1)}}{2} \geqslant \frac{s}{2}.$$

Therefore, we have

$$\sqrt{d'_{\text{tot}}} \geqslant \sqrt{\sum_{s=1}^{|R^*|} d'_{t_s}} \geqslant \sqrt{\sum_{s=1}^{|R^*|} \frac{s}{2}} = \sqrt{\frac{|R^*|(|R^*|+1)}{4}} \geqslant \frac{|R^*|}{2}.$$

**Part 2 (Bounding $d'_{\text{tot}}$):** Fix an arbitrary $Q \subseteq [T]$. Using $d'_t \leqslant \sqrt{d'_{\text{tot}}} + 1$ for all $t$, we have

$$\sum_{t \notin Q} d_t \geqslant \sum_{t \notin Q} d'_t \geqslant \max \left\{ 0, \ d'_{\text{tot}} - |Q|(\sqrt{d'_{\text{tot}}} + 1) \right\}.$$

Consequently, we can write

$$|Q| + \sqrt{\sum_{t \notin Q} d_t} \geqslant \min_{q \geqslant 0} \left\{ q + \sqrt{\max\{0, \ d'_{\text{tot}} - q(\sqrt{d'_{\text{tot}}} + 1)\}} \right\}$$

$$\geqslant \min \left\{ \sqrt{d'_{\text{tot}}} - 1, \ \min_{q \in [0, \sqrt{d'_{\text{tot}}} - 1]} \left\{ q + \sqrt{d'_{\text{tot}} - q(\sqrt{d'_{\text{tot}}} + 1)} \right\} \right\}$$

$$= \sqrt{d'_{\text{tot}}} - 1.$$

Combining both parts and minimizing over $Q \subseteq [T]$, we conclude

$$|R^*| + \sqrt{\sum_{t=1}^{T} d'_t} \leqslant 2\sqrt{d'_{\text{tot}}} + \sqrt{d'_{\text{tot}}} = 3\sqrt{d'_{\text{tot}}} = O\left( \min_{Q \subseteq [T]} \left\{ |Q| + \sqrt{\sum_{t \notin Q} d_t} \right\} \right),$$

completing the proof. $\square$

**Corollary 6.2** (Restated). *Under the conditions of Corollaries 5.3 and 5.7, applying the skipping wrapper yields:*

$$R_T(x^*) = O\left( GD\left[ \min_{Q \subseteq [T]} \left\{ |Q| + \sqrt{\sum_{t \notin Q} d_t} \right\} + \sqrt{T} \right] \right),$$

$$\bar{R}_T(x^*) = O\left( GD\left[ \min_{Q \subseteq [T]} \left\{ |Q| + \sqrt{\sum_{t \notin Q} d_t} \right\} + T^{\frac{3}{4}} \sqrt{\nu k} \right] \right),$$

*for $\mathcal{S}_{\text{ZS}}(\mathcal{W}_{\text{OCO}}(\mathcal{B}))$ and $\mathcal{S}_{\text{ZS}}(\mathcal{W}_{\text{BCO}}(\mathcal{B}))$, respectively.*

*Proof.* Follows immediately from Theorem 6.1 combined with Corollaries 5.3 and 5.7 for controlling $R'_T(x^*)$. $\square$

