# OpenReview forum: "A Reduction from Delayed to Immediate Feedback for Online Convex Optimization with Improved Guarantees"
_ICML.cc/2026/Conference — Submitted to ICML 2026_

### Official Review · Reviewer_gH8h · 2026-03-10

**Soundness:** 3
**Presentation:** 3
**Significance:** 3
**Originality:** 3
**Overall Recommendation:** 4
**Confidence:** 3

**Summary:**

This paper proposes a reduction-based framework for OCO with delayed feedback. Especially, for bandit convex optimization, the proposed methods significantly improved existing regret bounds.

**Compliance With Llm Reviewing Policy:**

Affirmed.

**Final Justification:**

The rebuttal have addressed my concerns and I will keep my score.

**Key Questions For Authors:**

see weaknesses

**Limitations:**

yes

**Strengths And Weaknesses:**

Strengths:

1. The paper is mostly clear. The studied problem is interesting and important for online learning area.
2. The theoretical contribution is solid. The authors propose a black-box reduction that reduces delayed OCO to online linear optimization with penalties for changing predictions. The proposed methods achieve improved regret bounds for bandit convex optimization.

Weaknesses:

1. The proposed method relies on the assumption of an oblivious adversary, which somewhat limits the generality and practical relevance of the results. In many real-world scenarios, adversaries may exhibit adaptive behavior based on the learner’s past actions. The authors may discuss the main technical difficulty on extending the framework to adaptive adversaries.

2. While the paper provides regret upper bounds for the proposed algorithm, the tightness of these bounds is not sufficiently discussed. It would be helpful if the authors could compare their results more explicitly with existing lower bounds or known optimal rates, and clarify whether the obtained bounds are optimal or if there is still room for improvement.

3. The formatting of the references appears somewhat informal. In particular, several entries in the reference list omit important bibliographic details such as the names of the conferences or journals where the cited works were published. Including complete and standardized citation information would improve the readability of the paper.

---

> ### Author Rebuttal · Authors · 2026-03-31
>
> We thank the reviewer for their detailed feedback. We will add a discussion of adversary models and lower bounds in the revised version, and we will also fix the reference formatting. We are also happy to report that our bounds are either tight or limited only by the best known efficient rates for the underlying non-delayed problem, with additional optimality results to appear in the revision.
> ## Weakness 1
> This is an important question. In delayed online learning with partial feedback (e.g., MAB or BCO), it is standard to assume that the delays are fixed in advance, since otherwise an adversary could strategically assign larger delays to rounds when the player is more likely to pick a better action. There is also relatively little work with adaptive losses in the delayed setting. A representative example is Bistritz et al. (2022), who study adaptive bandits with oblivious delays and argue that, in this regime, the right benchmark is “weighted regret” rather than standard regret. These difficulties are also applicable to our BCO setting.
>
> For OCO with first-order feedback, however, our algorithm performs deterministic updates, and the regret bound for every comparator $u \in \mathcal K$ holds pathwise for any realized sequence of losses and delays. Thus, our first-order results extend directly to adaptive losses or delays.
>
> For BCO, the situation is subtler because the learner relies on internal randomization to construct gradient estimates from partial feedback. In this regime, combining delay, bandit feedback, and adaptivity is significantly harder. We will add this discussion in the revised version to clarify why the oblivious-adversary is the standard in delayed bandit settings.
>
> ## Weakness 2
> Tightness in delayed OCO is governed by two distinct components, with a natural lower bound constructed as the maximum of the two: (1) a *delay-dependent* component, inherited from the delayed OCO with full-information feedback, and (2) a *delay-independent* component, inherited from the underlying non-delayed setting. Our delay-dependent terms are tight with respect to both *total delay* $d_{\mathrm{tot}}$ and *maximum number of outstanding observations* $\sigma_{\max}$, while the delay-independent terms are tight for first-order OCO and match the classical rates of Bandit Gradient Descent for BCO. We detail each component below.
> * *Delay-dependent component.* Lower bounds for delayed OCO with full-information feedback apply a fortiori to delayed first-order OCO or BCO. In the fixed-delay regime $d_t = d$, the optimal rates are already established: $\Theta(\sqrt{T d})$ under convexity and $\Theta(d \ln T)$ under strong convexity. Since $d_{\mathrm{tot}}=Td$ and $\sigma_{\max}=d$ in this regime, no guarantee expressed in terms of $d_{\mathrm{tot}}$ or $\sigma_{\max}$ can improve the $\sqrt{d_{\mathrm{tot}}}$ or $\sigma_{\max}\ln T$ delay terms. Our delay-dependent terms match these: $\sqrt{d_{\mathrm{tot}}}$ under convexity and $\min(\sqrt{d_{\mathrm{tot}}},\sigma_{\max}\ln T)$ under strong convexity.
> * *Delay-independent component.* For first-order OCO, the optimal non-delayed rates are $\Theta(\sqrt{T})$ and $\Theta(\ln T)$, both of which we match. For BCO, the $\Omega(k\sqrt{T})$ non-delayed BCO lower bound of Shamir (2013) has only been approached by methods with prohibitive computational costs (exponential or high-degree polynomial in $k, T$). Our framework builds on the classical single-point gradient estimator of Flaxman et al. (2004), whose Bandit Gradient Descent algorithm achieves $O(T^{3/4}k^{1/2})$ and $\tilde O(T^{2/3} k^{2/3})$ rates. Our reduction preserves these rates in the delayed setting.
> Since tight rates for non-delayed BCO itself remain open, closing the delayed BCO gap requires resolving the non-delayed one first. To demonstrate that our reduction framework is not the bottleneck, the revised version will include an extension to **delayed BCO with two-point feedback**  (Agarwal et al., 2010; Shamir, 2017). There, our guarantees are provably tight: $O(\sqrt{d_{\mathrm{tot}}}+\sqrt{Tk})$ and $O(\min(\sigma_{\max}\ln T,\sqrt{d_{\mathrm{tot}}})+k\ln T)$.
>
> ## Weakness 3
> Thank you for flagging this. We will revise the bibliography to include complete and standardized publication information.

---

> > ### Author Rebuttal · Reviewer_gH8h · 2026-04-01
> >
> > The rebuttal have addressed my concerns.

---

> > > ### Author Response · Authors · 2026-04-03
> > >
> > > Thank you for confirming that your concerns have been addressed. We are glad the rebuttal clarified that the oblivious adversary assumption is standard in delayed online learning and that our delay-dependent bounds are tight. Given this, we kindly ask you to consider whether your score of 4 still reflects your assessment, as it may weigh against the paper.

---

### Official Review · Reviewer_z6F4 · 2026-03-12

**Soundness:** 3
**Presentation:** 3
**Significance:** 3
**Originality:** 2
**Overall Recommendation:** 4
**Confidence:** 4

**Summary:**

The paper presents a unified reduction that converts a standard online convex optimization (OCO) algorithm into one capable of handling arbitrarily delayed feedback. The authors show that this reduction improves upon existing regret bounds for delayed OCO in both the first-order and bandit feedback settings.

**Compliance With Llm Reviewing Policy:**

Affirmed.

**Final Justification:**

The authors have addressed my concerns and I'll keep my score.

**Key Questions For Authors:**

1. Do the feedback signals arrive in the same order as the inputs? If not, how would the framework handle out-of-order feedback? Additionally, are the delay values assumed to be known to the learner?

2. Are there corresponding lower bounds for the delayed setting considered in this work?

3. Theorems 4.1 and 4.2 appear closely related to existing results in the literature on switching regret. For instance, similar bounds can be found in the work below and the references therein.

Reference

[1] Samrat Mukhopadhyay and Abhishek Sinha. “Online caching with optimal switching regret.” In Proceedings of the 2021 IEEE International Symposium on Information Theory (ISIT), pp. 1546–1551, 2021.

**Limitations:**

Yes

**Strengths And Weaknesses:**

$\textbf{Strengths}$

The paper is clearly written and easy to follow. The technical approach begins by introducing a continuous-time model that yields a regret decomposition into two components: a non-delayed term and a drift term. This decomposition is then used to reduce the delayed OCO problem to a standard OCO problem. The authors further extend the reduction to handle bandit feedback. The assumptions are clearly stated and the theoretical results appear rigorous. I particularly appreciated the dual perspective on delays and backlogs presented in the paper. See below for weaknesses.

---

> ### Author Rebuttal · Authors · 2026-03-31
>
> We thank the reviewer for their feedback and bringing the switching regret connection to our attention. We stress that in our setting observations can arrive in any order, and our algorithms are fully delay-adaptive — they can adapt to any delay sequence without prior knowledge of the delay structure. We are also happy to report that our bounds are either tight or limited only by the best known efficient rates for the underlying non-delayed problem, with additional optimality results to appear in the revision.
> ## Question 1
> No. Our framework allows observations to arrive in any order, not necessarily the prediction order. In the continuous-time model, this is captured by the observation-ordering permutation $\rho: [T] \to [T]$ (Definition 3.5), which sorts feedback by observation time and need not be the identity. Figure 3 even gives a concrete example for this.
>
> Our reduction handles this out-of-order feedback by updating a non-delayed base algorithm on observations in their order of arrival. By tuning the learning rates using delay-dependent quantities available at runtime, we obtain fully delay-adaptive guarantees.
>
> Importantly, we do **not** assume any prior knowledge of the delay structure at the start of the game. A key feasibility point of the reduction is that, at each observation time, the quantities required by the base algorithm can be computed from the history observed so far (Remark 5.1). This is precisely what enables the learning-rate schedules in our wrappers to be fully adaptive.
>
>
> ## Question 2
> Tightness in delayed OCO is governed by two distinct components, with a natural lower bound constructed as the maximum of the two: (1) a *delay-dependent* component, inherited from the delayed OCO with full-information feedback, and (2) a *delay-independent* component, inherited from the underlying non-delayed setting. Our delay-dependent terms are tight with respect to both *total delay* $d_{\mathrm{tot}}$ and *maximum number of outstanding observations* $\sigma_{\max}$, while the delay-independent terms are tight for first-order OCO and match the classical rates of Bandit Gradient Descent for BCO. We detail each component below.
> * *Delay-dependent component.* Lower bounds for delayed OCO with full-information feedback apply a fortiori to delayed first-order OCO or BCO. In the fixed-delay regime $d_t = d$, the optimal rates are already established: $\Theta(\sqrt{T d})$ under convexity and $\Theta(d \ln T)$ under strong convexity. Since $d_{\mathrm{tot}}=Td$ and $\sigma_{\max}=d$ in this regime, no guarantee expressed in terms of $d_{\mathrm{tot}}$ or $\sigma_{\max}$ can improve the $\sqrt{d_{\mathrm{tot}}}$ or $\sigma_{\max}\ln T$ delay terms. Our delay-dependent terms match these: $\sqrt{d_{\mathrm{tot}}}$ under convexity and $\min(\sqrt{d_{\mathrm{tot}}},\sigma_{\max}\ln T)$ under strong convexity.
> * *Delay-independent component.* For first-order OCO, the optimal non-delayed rates are $\Theta(\sqrt{T})$ and $\Theta(\ln T)$, both of which we match. For BCO, the $\Omega(k\sqrt{T})$ non-delayed BCO lower bound of Shamir (2013) has only been approached by methods with prohibitive computational costs (exponential or high-degree polynomial in $k, T$). Our framework builds on the classical single-point gradient estimator of Flaxman et al. (2004), whose Bandit Gradient Descent algorithm achieves $O(T^{3/4}k^{1/2})$ and $\tilde O(T^{2/3} k^{2/3})$ rates. Our reduction preserves these rates in the delayed setting.
> Since tight rates for non-delayed BCO itself remain open, closing the delayed BCO gap requires resolving the non-delayed one first. To demonstrate that our reduction framework is not the bottleneck, the revised version will include an extension to **delayed BCO with two-point feedback**  (Agarwal et al., 2010; Shamir, 2017). There, our guarantees are provably tight: $O(\sqrt{d_{\mathrm{tot}}}+\sqrt{Tk})$ and $O(\min(\sigma_{\max}\ln T,\sqrt{d_{\mathrm{tot}}})+k\ln T)$.
>
>
> ## Question 3
> We agree that making the connection between drift penalization and switching regret explicit is important. In particular, the drift term in Theorem 4.1 can indeed be viewed as a weighted switching penalty, where the weights are given by the dual-backlogs $\tilde{\sigma}_n^\star$. Thank you for pointing us to Mukhopadhyay and Sinha (2021). We will cite this connection in the revised version, while also clarifying that the main novelty in our setting is not the switching-type penalty itself, but the continuous-time reduction showing how delayed feedback induces this drift structure.

---

> > ### Author Rebuttal · Reviewer_z6F4 · 2026-04-01
> >
> > The authors have addressed my concerns.

---

> > > ### Author Response · Authors · 2026-04-03
> > >
> > > Thank you for confirming that your concerns have been addressed. We are glad the rebuttal highlighted that our fully delay-adaptive framework handles out-of-order feedback and requires no prior knowledge of the delay structure. Given this, we would be grateful if you could reconsider your score of 4, as with the current distribution a weak accept may weigh against the paper.

---

### Official Review · Reviewer_aput · 2026-03-13

**Soundness:** 4
**Presentation:** 4
**Significance:** 3
**Originality:** 4
**Overall Recommendation:** 5
**Confidence:** 4

**Summary:**

This paper studies delayed OCO and proposes a reduction method that converts delayed OCO to an online linear optimization problem. Based on this conversion, a wrapper is developed to convert any online linear optimization problem (with panelties for changing predictions) to a delayed OCO algorithm. By applying this framework to existing online algorithms, e.g., P-FTRL and OMD, this paper establishes new regret bounds for both first-order and bandit feedback and for both convex and strongly convex cases. With first-order feedback, this approach recovers state-of-the-art regret bounds. With bandit feedback, it improves state-of-the-art bounds.

**Compliance With Llm Reviewing Policy:**

Affirmed.

**Final Justification:**

All my comments have been addressed. I think this is a strong paper and would like to maintain my very positive score.

**Key Questions For Authors:**

- All bounds here are upper bounds for particular algorithms. To thoroughly understand the performance limit of delayed OCO, lower bounds on any algorithm may be very useful. If the regret bounds of the proposed algorithms match the lower bounds, "optimality" of the algorithm can be established.
- Though the paper is mostly theoretical, some experiments may be helpful to validate the practical merit of the proposed algorithms.

**Strengths And Weaknesses:**

Strength:
- Comprehensive and unified approach for handling delayed OCO. The development of the reduction and wrapper approach are interesting.
- Can recover/improve the state-of-the-art regret bounds for delayed OCO and the analysis is unified, the framework is general.

Weakness:
- All bounds here are upper bounds for particular algorithms. To thoroughly understand the performance limit of delayed OCO, lower bounds on any algorithm may be very useful. If the regret bounds of the proposed algorithms match the lower bounds, "optimality" of the algorithm can be established.
- Though the paper is mostly theoretical, some experiments may be helpful to validate the practical merit of the proposed algorithms.

---

> ### Author Rebuttal · Authors · 2026-03-31
>
> We thank the reviewer for their detailed feedback and for recognizing the generality of our framework. We are happy to report that our bounds are either tight or limited only by the best known efficient rates for the underlying non-delayed problem, with additional optimality results to appear in the revision. We address each question in turn.
> ## Question 1
> Tightness in delayed OCO is governed by two distinct components, with a natural lower bound constructed as the maximum of the two: (1) a *delay-dependent* component, inherited from the delayed OCO with full-information feedback, and (2) a *delay-independent* component, inherited from the underlying non-delayed setting. Our delay-dependent terms are tight with respect to both *total delay* $d_{\mathrm{tot}}$ and *maximum number of outstanding observations* $\sigma_{\max}$, while the delay-independent terms are tight for first-order OCO and match the classical rates of Bandit Gradient Descent for BCO. We detail each component below.
> * *Delay-dependent component.* Lower bounds for delayed OCO with full-information feedback apply a fortiori to delayed first-order OCO or BCO. In the fixed-delay regime $d_t = d$, the optimal rates are already established: $\Theta(\sqrt{T d})$ under convexity and $\Theta(d \ln T)$ under strong convexity. Since $d_{\mathrm{tot}}=Td$ and $\sigma_{\max}=d$ in this regime, no guarantee expressed in terms of $d_{\mathrm{tot}}$ or $\sigma_{\max}$ can improve the $\sqrt{d_{\mathrm{tot}}}$ or $\sigma_{\max}\ln T$ delay terms. Our delay-dependent terms match these: $\sqrt{d_{\mathrm{tot}}}$ under convexity and $\min(\sqrt{d_{\mathrm{tot}}},\sigma_{\max}\ln T)$ under strong convexity.
> * *Delay-independent component.* For first-order OCO, the optimal non-delayed rates are $\Theta(\sqrt{T})$ and $\Theta(\ln T)$, both of which we match. For BCO, the $\Omega(k\sqrt{T})$ non-delayed BCO lower bound of Shamir (2013) has only been approached by methods with prohibitive computational costs (exponential or high-degree polynomial in $k, T$). Our framework builds on the classical single-point gradient estimator of Flaxman et al. (2004), whose Bandit Gradient Descent algorithm achieves $O(T^{3/4}k^{1/2})$ and $\tilde O(T^{2/3} k^{2/3})$ rates. Our reduction preserves these rates in the delayed setting.
> Since tight rates for non-delayed BCO itself remain open, closing the delayed BCO gap requires resolving the non-delayed one first. To demonstrate that our reduction framework is not the bottleneck, the revised version will include an extension to **delayed BCO with two-point feedback**  (Agarwal et al., 2010; Shamir, 2017). There, our guarantees are provably tight: $O(\sqrt{d_{\mathrm{tot}}}+\sqrt{Tk})$ and $O(\min(\sigma_{\max}\ln T,\sqrt{d_{\mathrm{tot}}})+k\ln T)$.
> ## Question 2
> We will include experiments comparing our delayed BCO algorithm against Wan et al. (2024) under queueing-motivated delays. Losses are drawn i.i.d., and the delay sequence is generated by a continuous-time $M/G/\infty$ queueing model —  prediction times $l_t$ follow a Poisson process, while the service times $r_t - l_t$ are sampled from the service-time distribution $G$.   To our knowledge, this is the first experimental delayed OCO setup to produce realistic *correlated* delay sequences. Prior experimental work has been limited to fixed or i.i.d. stochastic delays; we believe this queueing-based methodology further highlights the importance of our Continuous Time Model.
>
> With heavy-tailed service-time distributions $G$, this produces delay sequences where $d_\max$​ far exceeds average delay, precisely the regime where our algorithm should outperform the $d_{\max}$-dependent guarantees of Wan et al. We will plot cumulative regret versus $T$ across varying tail heaviness, and expect our regret to remain stable as the tail grows heavier while theirs rapidly degrades.

---

> > ### Author Rebuttal · Reviewer_aput · 2026-04-01
> >
> > All my comments have been addressed.

---

### Official Review · Reviewer_QC2c · 2026-03-20

**Soundness:** 3
**Presentation:** 3
**Significance:** 3
**Originality:** 3
**Overall Recommendation:** 5
**Confidence:** 3

**Summary:**

This paper studies a unified model for the reduction of online convex optimization under delay and two feedback models (first-order and bandits). To achieve a unified framework for this reduction, the authors first introduce a continuous-time model that serves as one of the key steps toward unlocking such a reduction and achieving a unified framework for the two feedback models. Then, the authors present the theoretical results for both feedback models (first-order and bandits). For the bandits model, their results improve the delay-induced regret over the state-of-the-art results. For the first-order feedback, the result matches the state-of-the-art, but with simpler analysis.

**Compliance With Llm Reviewing Policy:**

Affirmed.

**Key Questions For Authors:**

Please answer the questions posed in the weaknesses section above.

**Limitations:**

The paper overall is in good shape, and it could benefit from a better clarification of the significance and technical novelties of the results.

**Strengths And Weaknesses:**

**Strengths**

-- The paper is very well-written with a clear positioning of the paper in the literature

-- The execution of the results from introducing the continuous-time model to the two feedback model is elegant, clearly presented, and shows a very logical flow.

-- The results are solid, significant, and clearly advance the state-of-the-art results in the literature, on both theoretical bounds and the technical novelties to get there.

**Weaknesses**

-- While the theoretical results sound solid (I did not check the details), the theoretical parts of the paper are pretty dense and do not provide sufficient remarks and insight on (1) how the results are concretely comparable with the most relevant results in the literature (I mean in a more fine-grained details than what is in Table 1, on, e.g., how siginificants are the improvements, and what barriers are broken in prior work to unloch such improvements); and (2) what is the technical novelties to achieve such results that are not present in relevant prior work.

---

> ### Author Rebuttal · Authors · 2026-03-31
>
> We thank the reviewer for their positive evaluation and address their request for finer-grained comparisons and technical novelty below. We will incorporate these details into the Technical Contributions paragraph of the introduction.
> ## Question 1 - improvements compared to previous work.
> Our delayed BCO bounds strictly improve on all prior work, are the first to be provably tight in the delay-dependent terms, and are achieved by fully adaptive algorithms that require no advance knowledge of the delay structure.
>
> *Strictly sharper rates.* Our convex delay term $\sqrt{d_{\text{tot}}}$ improves on $(Td_{\text{tot}})^{1/3}$ of Bistritz et al. (2022) for all delay sequences, since $d_{\text{tot}} \le T^2$. For example, under fixed delays $d_t = T^{1/2}$, their regret scales as $T^{5/6}$ while ours scales as $T^{3/4}$. Against Wan et al. (2024), the gap is more severe: their guarantees scale with $d_{\max}$ and require advance knowledge of both $d_{\max}$ and $T$. If $d_1 = \Theta(T)$ and $d_t = 0$ for $t > 1$, their bounds become linear; ours remain sublinear: $O(T^{3/4}k^ {1/2})$ and $\tilde{O}(T^{2/3}k^{2/3})$.
>
> *Skipping wrapper.* Our skipping wrapper further sharpens the delay contribution from $O(\sqrt{d_{\text{tot}}})$ to $O (\min_{Q \subseteq [T]} |Q| + \sqrt{\sum_{t \notin Q} d_t})$, which can be substantially smaller when only a few rounds incur large delays. For instance, if the first $\Theta(T^{3/4})$ rounds have delay $\Theta(T)$ and the rest have no delay, the delay-dependent term improves from $O(T^{7/8})$ to $O(T^{3/4})$.
>
> *Tightness.* Our delay-dependent terms are tight with respect to both total delay $d_{\text{tot}}$ and maximum number of outstanding observations $\sigma_\max$. We are the first to establish this for delayed BCO, matching the optimal rates known for first-order OCO. For the delay-independent term, we match the classical rate of Bandit Gradient Descent (Flaxman et al., 2004).
>
> *Extension to BCO with two-point feedback.*  In the revised version we include Delayed BCO with two-point feedback (as in Agarwal et al., 2010; Shamir, 2017), providing the first known rates for this setting, which are also tight: $O(\sqrt{d_{\text{tot}}}+\sqrt{Tk})$ and $O(\min(\sigma_{\max}\ln T,\sqrt{d_{\text{tot}}})+k\ln T)$.
> ## Question 2 - technical novelties.
> The core difficulty of delayed BCO, compared to first-order delayed OCO, is that gradient-based algorithms must rely on high-variance single-point gradient estimates, making prediction drift much harder to control. Our principal tool for addressing this is the Continuous Time Model (CTM): by shifting from a prediction-centric to an observation-centric viewpoint, it enables a delay-adaptive variance reduction (Theorem 5.6 and Lemma 4.2) that controls prediction drift without collapsing the delay structure. Novel P-FTRL drift-control lemmas (Lemmas 4.2 and D.4) are vital to the analysis of this variance reduction.
>
> Prior approaches are fundamentally more lossy. Wan et al. (2024) reduce variance by batching gradient estimates into fixed-size blocks, but this forces all per-round delays to be upper-bounded by $⁡d_{\max}$​, discarding finer delay structure and making their guarantees $⁡d_{\max}$​-dependent. Bistritz et al. (2022) do not perform variance reduction at all, relying on single-step OMD-style drift control, which is why their delay term remains at $(Td_{\text{tot}})^{1/3}$. Running P-FTRL as a base algorithm in our reduction moves this down to $\sqrt{d_{\text{tot}}}$ (and $\min(\sigma_{\max}\ln T,\sqrt{d_{\text{tot}}})$ under strong convexity), matching the optimal rates for delayed first-order OCO.​

---

> > ### Author Rebuttal · Reviewer_QC2c · 2026-04-01
> >
> > The authors have addressed my concerns.

---

### Official Review · Reviewer_7h5x · 2026-03-20

**Soundness:** 3
**Presentation:** 3
**Significance:** 2
**Originality:** 2
**Overall Recommendation:** 3
**Confidence:** 4

**Summary:**

This paper studies the problem of online convex optimization (OCO) with arbitrary delays, which has attracted much interest recently. A reduction-based framework is proposed and used to recover best existing results in the full-information setting, and achieve improved results in the bandit setting, i.e., bandit convex optimization (BCO). Specifically, for BCO with convex functions, this paper achieves $O(\sqrt{d_{tot}}+T^{3/4}\sqrt{k})$ regret bound, which is better than the existing $O(\sqrt{d_{max}T}+T^{3/4}\sqrt{k})$ and $O((Td_{dot})^{1/3}+T^{3/4}\sqrt{k})$ regret bounds. For BCO with strongly convex functions, this paper achieves $O(\min\{\sigma_{max}\ln T, \sqrt{d_{dot}}\}+(T^2\ln T)^{1/3}k^{2/3})$ regret bound, which is better than the existing $O(d_{max}\ln T+(T^2\ln T)^{1/3}k^{2/3})$ regret bound.

**Compliance With Llm Reviewing Policy:**

Affirmed.

**Final Justification:**

After the rebuttal and the discussion with the authors, I am still not very convinced by the necessity of the proposed framework and the CTM model.

Compared this paper with existing studies, the main improvements of this paper are the bounds for BCO. My major concern is that these bounds can be achieved in a much simpler way. In the initial rebuttal, the authors argue that this is incorrect. However, after the further discussion, the authors agree that my proposal can address the convex case, but still argue that their strongly convex bound cannot be recovered. Actually, I also believe that, based on my proposal, it is also easy to achieve the strongly convex bound by incorporating the analysis in Qiu et al. (2025). This still does not rely on the proposed framework and the CTM model.

Overall, I do not think that a reduction is essentially more significant than concrete algorithms, especially when it does not seem more elegant. Moreover, I do not even agree that the proposed framework is a **black-box** reduction, because combining it with the algorithms enjoying almost the same guarantees for non-delayed OCO or BCO can result in obviously different bounds for the delayed setting.

**Key Questions For Authors:**

1) Please provide a more detailed comparison between the proposed reduction and the existing one in Joulani et al. (2016).

2) If the reduction-based framework is combined with bandit gradient descent, what is the final regret bound?

3) Even for the full-information setting with strongly convex functions, the final regret bounds achieved by combining the reduction-based framework with P-FTRL and OMD are different. To some extent, this difference weakens the generality of the reduction-based framework, i.e., one may prefer to design the delayed variant of a specifical algorithm.

**Limitations:**

Yes

**Strengths And Weaknesses:**

This paper is well-written in the sense that the technical details are easy to follow. Although I cannot check all the proofs, it is not surprising to see the derived results. Moreover, the improvements to the bandit setting are interesting. However, there are still some concerns on the technical novelty and significance.
1) The proposed reduction-based framework actually is very similar to the unified framework of Joulani et al. (2016). The essence of these two frameworks is that if one traditional OCO algorithm is run over the loss functions in the observed sequence, the regret of delayed OCO can be decomposed into the non-delayed regret of this algorithm and a prediction drift term (see Theorem 1 of Joulani et al. (2016) for checking the similarity). Moreover, such a reduction actually has been implicitly used in some existing delayed algorithms (e.g., Eq. (10) in [1], Eq. (11) in [2], and Eq. (5) in [3]).
2) In the proposed reduction-based framework, a continuous time model (CTM) is introduced. Although some properties of such a model may be useful for the analysis, I do not think that the so-called "continuous" model is necessary. Actually, I believe similar properties have been used in the literature, but only for one-to-one match of an original round $t$ and the observation round (or other related rounds).
3) It seems that both the reduction-based framework and the CTM model are not necessary for the improvements on BCO. In contrast, these improvements should owe to a tighter expected upper bound on outstanding estimated gradients (see Lemma D.4). By using Lemma D.4, it is not hard to show that the algorithm in Wan et al. (2024) can also achieve (or nearly) these improved regret bounds if its block size is set to be $1$. Actually, except for the adjustment of $\eta_t$, the derived BCO algorithm in this paper is the same as the algorithm in Wan et al. (2024) with the block size $1$. Moreover, notice that when combining with bandit gradient descent---another classical BCO algorithm, the reduction-based framework does not achieve these improvements.

[1] Wan et al. Non-stationary Online Convex Optimization with Arbitrary Delays. In ICML 2024.

[2] Wan et al. Projection-free Online Learning with Arbitrary Delays. In arXiv:2204.04964v2.

[3] Hsieh et al. Multi-Agent Online Optimization with Delays: Asynchronicity, Adaptivity, and Optimism. In JMLR 2022.

---

> ### Author Rebuttal · Authors · 2026-03-31
>
> We thank the reviewer for their comments. We will cite the mentioned papers and expand the comparison (appears in Introduction) to Joulani et al. (2016). That said, we stress that our improved bounds for delayed BCO do not follow from any previous work. Achieving them required new non-trivial ideas — in particular, the Continuous Time Model and its structural invariants — which we believe are of independent interest and applicable to any delayed online learning setting.
> ## Weakness 1 / Question 1
> The idea of updating parameters upon each observation is foundational in asynchronous optimization, dating back to Tsitsiklis (1984). Joulani et al. leveraged it for delayed OCO with full-information feedback, but without formalizing the observation ordering or identifying any structural invariants, they yield weaker, non-adaptive bounds that require advance knowledge of the delay structure to set learning rates. They provide no tools for controlling prediction drift when gradients are replaced by high-variance bandit estimates, which is the central challenge in delayed BCO. Our work excels in the following concrete ways:
> 1. *We achieve delay-adaptivity.* The CTM formalizes observation-ordering via the permutation $\rho:[T]\to[T]$ and establishes structural invariants (Theorem 3.4) that enable fully delay-adaptive learning rates. We also clarify a subtle issue: their $O(\sqrt{T\tau^\star})$ bound is in terms of $\tau^\star$, which they equate with $d_\max$, but $\tau^\star$ in fact corresponds to our dual-delay $d^\star_\max$, a distinct quantity (see Figure 2). Theorem 3.4(b) shows $d^\star_\max=\Theta(d_\max)$, an identification absent from prior work.
> 2. *We extend the scope.* Joulani et al. focus only on full-information OCO. We cover both first-order OCO and BCO. Moreover, in the revision we include delayed BCO with two-point feedback, providing the first known rates: $O(\sqrt{d_{\text{tot}}}+\sqrt{Tk})$ and $O(\min(\sigma_{\max}\ln T,\sqrt{d_{\text{tot}}})+k\ln T)$.
> ## Weakness 2
> The CTM is the principal idea at the core of our reduction: it enables a shift from a prediction-centric to an observation-centric viewpoint. It yields an exact identity (Theorem 3.4(c)) for the number of observations between prediction in round $t$ and observation in $t+d_t$, a quantity that prior work (e.g., Thune et al., 2019) could only bound by $O(d_\max)$. While delay-adaptivity for full-info OCO has been achieved in discrete models, delayed BCO requires controlling prediction drift under high-variance gradient estimates. The CTM accomplishes this via a delay-adaptive variance reduction (Theorem 5.6, Lemmas 4.2/D.4). This is why ours is the first delay-adaptive algorithm for delayed BCO with these improved rates.
> ## Weakness 3 / Question 2
> The reviewer suggests that our BCO improvements can be recovered without the CTM by setting the block size in Wan et al. (2024) to 1. This is incorrect — no previous method achieves these bounds, and no such claim appears in the literature. With block size 1, blocking disappears entirely and their method reduces to delayed FTRL of Flaspohler et al. (2021) applied to gradient estimates. Our algorithm is fundamentally different: it performs single-instance P-FTRL updates in order of feedback arrival, and it is the CTM that makes analyzing such updates and deriving delay-adaptive guarantees possible.
>
> Beyond the algorithmic distinction, Wan et al. are limited by their $⁡d_{\max}$​-dependence. They handle round-dependent delays only by upper-bounding them with $d_\max$ (Lemma 6), collapsing the problem to the worst-case fixed-delay regime ($\forall t\in[T],d_t=d_\max$). They themselves acknowledge this as an open problem. The gap is substantial: with a single large delay $d_1=T-1$ and $d_t=0$ for $t>1$, their bounds are linear while ours are sublinear: $O(T^{3/4})$ and $\tilde{O}(T^{2/3})$.
>
> Regarding the base algorithm: the regret bounds provided by our reduction depend on the base algorithm (Theorem 5.6), as is inherent to any black-box reduction. The improved delayed BCO rates in Corollaries 5.7/5.8 rely on the novel P-FTRL drift bound (Lemmas 4.2/C.6); the OMD drift control (Theorem 4.1) yields weaker guarantees. We interpret the reviewer's question about BGD as about OMD base algorithm: one can match Bistritz et al. (2022) when $d_{\text{tot}}\ge T^{5/4}$, and achieve $\tilde O(\sigma_\max T^{1/3}+T^{2/3})$ under strong convexity — improving on Wan et al. when $\sigma_\max\le d_\max/T^{1/3}$.
>
> ## Question 3
> The difference between the P-FTRL and OMD bounds in Corollary 5.4 does not weaken the generality of the reduction. As in Q2, the gap stems from different drift-control properties of base algorithms (Lemma 4.2 vs Theorem 4.1), and our framework propagates this distinction to the delayed setting, as a black-box reduction should. Since $\sigma_\max\le\sqrt{2d_{\text{tot}}}$, the two bounds differ by at most a log-factor, and both yield an improvement over $O(d_\max\ln T)$ of Wan et al.

---

> > ### Author Rebuttal · Reviewer_7h5x · 2026-04-02
> >
> > Thanks for the authors' response. However, my major concerns are not addressed. First, I notice that the reduction of Joulani et al. (2016) focuses on the full-information feedback, i.e., the loss function $f_t(\cdot)$. However, from the previous studies (I suggested in the initial comments), it can be easily extended into the setting with only gradient or bandit feedback.
> >
> > Moreover, I am still not very convinced by the necessity of the proposed framework and the CTM model. Compared with existing studies, the main improvements of this paper are the bounds for BCO. However, I believe that these bounds can be achieved in a much simpler way, i.e., the algorithm of Wan et al. (2024) with the block size of $1$. Since the authors hold the opposite, I provide a brief explanation on how to achieve this.
> > 1) By substituting the block size of $1$ into Lemma 2 and Eq. (17) of Wan et al. (2024), we then only need to bound the following term
> > $$
> > \sum_{t=1}^T\\|y_t-y_t^\ast\\|_2
> > $$
> > where the notations $y_t$ and $y_t^\ast$ follow Wan et al. (2024).
> > 2) From Eq. (19) of Wan et al. (2024), we notice that the term $\|y_t-y_t^\ast\|_2$ can be further bounded by
> > \\[
> > \\frac{\\eta}{2}\\left\\|\\sum\_{k\\in U\_t} g\_k\\right\\|\_2
> > \\]
> > where $\eta$ is a parameter, $g_k$ is the estimated gradient of round $k$, and the set $U_t$ consists of the time stamp of loss values that are queried but still not arrive at the end of round $t-1$.
> > 3) A direct upper bound of $\\|\\sum\_{k\in U\_t} g\_k\\|\_2$ is $O(|U_t|/\delta)$, where $\delta$ is the exploration radius of estimated gradient. One may notice that such a bound will only lead to the result of Bistritz et al. (2022). Fortunately, by following the proof of Lemma D.4, we actually can improve the expected upper bound of $\\|\\sum\_{k\in U\_t} g\_k\\|\_2$ to $O(|U_t|+1/(\delta^2))$. Note that the key to this improvement is to introduce the conditional expectation of each $g_k$. The reduction framework and the CTM model are not required.
> > 4) Finally, combining the above results with the well-known fact that $\sum_{t=1}^T|U_t|\leq d_{tot}$, and tuning the parameter $\eta$ according to $\sum_{t=1}^T|U_t|$, it is sufficient to recover the improved regret bounds for BCO. Note that although $\sum_{t=1}^T|U_t|$ is unknown beforehand, it can be estimated by using the doubling trick.

---

> > > ### Author Response · Authors · 2026-04-03
> > >
> > > ## On the alternative algorithm derived from our analysis.
> > >
> > > **Let us summarize the new algorithm you suggest that uses parts of our analysis.**
> > > You take the blocking algorithm of Wan et al. (2024) with block size 1 — which is equivalent to the delayed FTRL of Flaspohler et al. (2021) applied to bandit gradient estimates — modify learning rates via a doubling trick over $d_{\mathrm{tot}}$, effectively matching the adaptive learning rates in our algorithm, and apply **our** Lemma D.4 in the analysis. You claim this should match our rates for delayed BCO with convex losses (our Corollary 5.7). We acknowledge this may be possible, though we have not verified the details. We disagree, however, that this diminishes the contribution of our paper, for the following reasons.
> > >
> > > 1. Your analysis relies on **our** Lemma D.4 and therefore cannot be derived from previous work alone.
> > > 2. Your proposal (based on our analysis) addresses only the convex case and **does not recover our strongly convex bound** $\min(\sigma_{\max} \ln T, \sqrt{d_{\mathrm{tot}}})$ (Corollary 5.8), which arises from two complementary drift-control mechanisms unified through Theorem 3.4 and Fact B.4. Our algorithm obtains improved rates for both convex and strongly convex losses without any unnecessary doubling tricks over both $d_{\text{tot}}$ and $T$.
> > > 3. Our work provides a **black-box reduction** from delayed to non-delayed online learning — a substantially stronger contribution than a single algorithm tailored to one specific setting.
> > >
> > >
> > > ## The reduction framework and the CTM are our contributions.
> > >
> > > The core contribution of our paper is a **black-box reduction** from delayed to non-delayed online learning — we do not merely improve delayed BCO rates, we do so through a general-purpose framework that applies to any base algorithm and both feedback models.
> > >
> > > The **CTM plays an essential role in enabling this generality**, and both the model and the results we prove about it are of independent interest. Real systems generate predictions and observations asynchronously — the standard round-based model is fundamentally weaker. The CTM captures this directly, and the structural results it yields — dual-delays, dual-backlogs, their invariants (Theorem 3.4), and the observation-ordering framework — are novel and applicable to **any** delayed online learning setting. We believe they are closely connected to problems arising in asynchronous distributed optimization, queueing theory, and online scheduling.

---

### Decision · Program_Chairs · 2026-04-30

**Decision:**

Reject

**Comment:**

This paper develops a reduction-based framework for online convex optimization with delayed feedback, centered around a continuous-time model and a unified treatment of first-order and bandit feedback. The paper is technically interesting, and the improved delayed BCO bounds are a potentially meaningful part of the submission.

The main unresolved issue is about the source of the paper’s contribution: in particular, whether the improved BCO guarantees fundamentally rely on the proposed reduction/continuous-time framework, or whether they can in essence be obtained by a simpler adaptation of prior work together with part of the new analysis. Reviewer 7h5x articulated this concern in a concrete way, and the follow-up discussion did not fully resolve it. Given this remaining ambiguity, it is difficult to confidently assess how much of the advance should be attributed to the framework versus to the improved BCO analysis. For that reason, the current version does not yet make a sufficiently strong case for acceptance. The paper does appear promising, and a careful revision that makes much more explicit where the performance improvements come from, what role the CTM/reduction framework is truly playing, and which parts of the contribution remain even if comparable BCO bounds could be derived more directly would likely strengthen it substantially.